# ADVERSARIAL ROBUSTNESS OF CONTINUOUS TIME DYNAMIC GRAPHS

## ABSTRACT

Real-world relations are dynamic and often modeled as temporal graphs, making Temporal Graph Neural Networks (TGNNs) crucial for applications like fraud detection, cybersecurity, and social network analysis. However, our study reveals critical vulnerabilities in these models through three types of adversarial attacks: structural, contextual, and temporal perturbations. We introduce Temporally-aware Randomized Block Coordinate Descent (TR-BCD), a novel gradient-based evasion attack framework for continuous-time dynamic graphs. Unlike previous approaches that rely on heuristics or require training data access, TR-BCD optimizes adversarial edge selection through continuous relaxation while maintaining realistic temporal patterns. Through extensive experiments on six temporal networks, we demonstrate that TGNNs are highly vulnerable to TR-BCD attacks, reducing Mean Reciprocal Rank (MRR) by up to 53% while perturbing only 5% of edges. Our attacks are highly effective against state-of-the-art models, including TGN and TNCN, highlighting the importance of studying adversarial robustness for temporal graph learning methods.

## 1 INTRODUCTION

Graphs are ubiquitous in representing complex relations between entities, such as social networks (Kumar et al., 2019), traffic networks (Ding et al., 2021), biological networks (Barabasi & Oltvai, 2004), transaction networks (Ni et al., 2019; Shamsi et al., 2022) and political networks (Fowler, 2006; Huang et al., 2020). Recently, Graph Neural Networks (GNNs) have demonstrated state-of-the-art performance across a variety of graph learning tasks (Hu et al., 2020; 2021). However, there has been compelling evidence showing that GNNs are not robust to adversarial perturbations (Zügner et al., 2018; Günnemann, 2022; Ma et al., 2020), which raises concerns about their deployment in real-world large-scale applications (Hamilton et al., 2017; Ying et al., 2018).

Many real-world graphs are inherently dynamic, with frequent node or edge additions. These evolving networks are often modeled as *temporal graphs*, requiring ML models to learn both structural and temporal dependencies. To tackle this challenge, Temporal Graph Neural Networks (TGNNs) have been proposed to perform tasks such as link prediction (Huang et al., 2024) and node classification (Rossi et al., 2020). One popular approach is the Temporal Graph Network (TGN) (Rossi et al., 2020), which processes a continuous stream of edges (in the continuous-time dynamic graph setting) and makes predictions for future events based on past interactions stored in *memory*.

While the robustness of static graphs has been extensively studied (Xu et al., 2019; Zügner et al., 2018; 2020; Wang & Gong, 2019; Wu et al., 2019; Dai et al., 2018), the vulnerabilities of TGNNs to adversarial perturbations remain under-explored. The increasing deployment of temporal graph learning methods in high-stakes applications makes understanding their robustness critical. For example, in financial transaction networks, fraudsters can strategically insert fake transactions to evade detection systems - a single compromised account could be used to create seemingly legitimate transaction patterns that mask fraudulent activity (Kim et al., 2024). Similarly, in cybersecurity, attackers can carefully time network connections and craft traffic patterns to avoid intrusion detection systems that rely on temporal graph analysis (Idé & Kashima, 2004; Yoon et al., 2019). Even in social networks, malicious actors can orchestrate coordinated influence campaigns by tactically building connections over time to maximize their reach while appearing organic to automated detection methods (Del Vicario et al., 2016). These scenarios highlight how adversaries can exploit

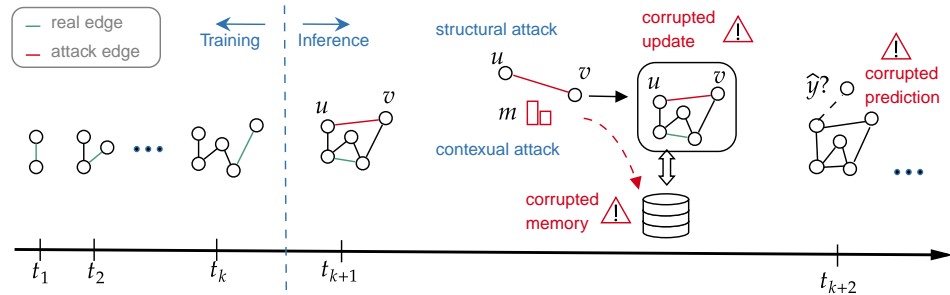

Figure 1: Evasion attacks for CTDGs. Both *structural* and *contextual* attacks are applied at inference time on a trained model. The adversarial attacks corrupts the model memory.

temporal patterns to circumvent safeguards. A key advantage of TGNNs is their ability to rapidly update internal node representations during inference without requiring backpropagation (Rossi et al., 2020). In contrast to prior work focusing on poisoning attacks during training (Lee et al., 2024), we investigate adversarial evasion attacks—attacks occurring after model deployment that do not require modifications to the training data (see Figure 1). While the only existing evasion attack on TGNNs (Dai et al., 2023) is limited in scope, we formalize a general attack setting and propose a gradient-based method that inserts new edges among existing nodes.

Temporal graphs present unique challenges due to the interplay between temporal, structural, and contextual dimensions. In static graphs, adversarial modifications are confined to the topology; however, temporal graphs involve an additional attack surface through the timing and sequencing of interactions. These graphs encapsulate multiple layers of information: namely, structure, attributes, and temporal evolution, which necessitates adversarial attacks that can modify graph topology, corrupt edge features, and alter chronological order. Consequently, perturbations must be precisely timed and coordinated with the graph's natural evolution while navigating a vast combinatorial search space. This temporal attack setting presents three interconnected technical challenges. *First, memory complexity challenge*: naïvely optimizing edge perturbations requires $\Theta(|E| \times |V|^2)$ memory to store all possible edge perturbations, prohibitive for large graphs (e.g., Reddit: 243GB). *Second, temporal propagation challenge*: unlike static graphs where perturbations are isolated, TGNNs' memory modules cause corrupted states to persist and compound across future predictions. *Third, candidate selection challenge*: with budget constraints, randomly sampling from $|V|^2$ possible edges is suboptimal; we need to exploit TGNN training objectives to identify high-impact candidates. To address these challenges, we introduce three types of perturbations for Continuous-Time Dynamic Graphs (CTDGs): (1) *structural perturbations* through the insertion of adversarial edges, (2) *contextual perturbations* that corrupt edge features in attributed graphs, and (3) *temporal perturbations* that adjust timestamps for adversarial edge insertions to maintain temporal consistency. These perturbations can be applied individually or in combination. To incorporate these perturbations, we introduce Temporally-aware Randomized Block Coordinate Descent (TR-BCD), the first gradient-based evasion attack strategy for TGNNs to explore these attacks. Unlike existing methods that rely on heuristic constraints and require training data access (Lee et al., 2024), or use heuristic attack strategies with simplistic evaluation metrics (Dai et al., 2023), TR-BCD operates as an evasion attack optimizing adversarial edge selection through continuous relaxation, taking the model architecture and parameters into account. We evaluated TR-BCD against Temporal Graph Benchmark (TGB) (Huang et al., 2024), which frames link prediction as a ranking problem using Mean Reciprocal Rank (MRR) over historical and random negatives, ensuring a robust evaluation of attack effectiveness. Our contributions are summarized as follows:

- **General framework for adversarial attack on CTDGs.** In this work, we present a general framework of adversarial attacks on CTDGs, including three complementary perturbation types: structural (edge insertions), contextual (feature modifications), and temporal (timestamp manipulation). Our framework defines permissible perturbations through principled constraints on budget, temporal patterns, and feature deviations to ensure realistic attacks.
- **Novel Temporally-aware evasion attack.** We propose TR-BCD, a gradient-based method that strategically distributes adversarial edges across temporal batches. TR-BCD optimizes edge selection through continuous relaxation and leverages both random and challenging historical

negative samples as attack candidates. Unlike previous work that relies on heuristics or requires training data access, our method directly optimizes an adversarial objective during inference.

- **Extensive empirical evaluation revealing vulnerabilities.** Through comprehensive experiments across six datasets, we demonstrate that TR-BCD consistently degrades model performance, achieving up to 53% reduction in Mean Reciprocal Rank (MRR) while perturbing only 5% of edges. Our attacks are highly effective against state-of-the-art models including TGN and TNCN, revealing critical vulnerabilities in current temporal graph learning approaches.

## 2 RELATED WORK

**Temporal Graph Learning.** Kazemi et al. (2020) categorized temporal graphs into Discrete Time Dynamic Graphs (DTDGs) and Continuous Time Dynamic Graphs (CTDGs). In this work, we focus on CTDGs, however our adversarial attack formulation in Section 3 can be easily extended to DTDGs. CTDG methods receive a continuous stream of edges as input and make predictions over any possible timestamps. For efficiency, the stream is typically divided into fixed-size batches processed sequentially. CTDG methods incorporate newly observed information by updating internal representations, often tracking node states over time and sampling temporal neighborhoods for prediction. Temporal Graph Network (TGN) (Rossi et al., 2020) introduced a memory-based encoder architecture that produces node embeddings for downstream tasks like link prediction and node classification. Its memory module stores node histories to model long-term dependencies, aggregating embeddings of participating nodes and their temporal neighbors for predictions. Building on this, Temporal Neural Common Neighbor (TNCN) (Zhang et al., 2024) enhanced link prediction by incorporating common neighbors into more discriminative edge representations through a temporal dictionary of multi-hop neighbors, achieving state-of-the-art performance.

**Robustness of GNNs and TGNNs.** RL-S2V (Dai et al., 2018) and Nettack (Zügner et al., 2018) pioneered adversarial attacks on node classification by manipulating both graph structure and node features. Similar to Nettack, Projected Gradient Descent (PGD) (Xu et al., 2019) introduced general gradient-based topology attacks through iterative optimization of edge perturbation matrices. However, PGD's memory requirements scale quadratically with the number of nodes. Projected Randomized Block Coordinate Descent (PR-BCD) (Geisler et al., 2021) improves scalability with sparsity-aware optimization that iteratively generates sparse adjacency matrices while satisfying budget constraints. Its greedy variant, GR-BCD (Geisler et al., 2021), which we adapt for our work, efficiently selects optimal source-node pairs for adversarial edge insertion. MemStranding (Dai et al., 2023) attacks temporal graph networks by corrupting node memories through strategically injected fake events. It identifies high-degree victim nodes and their neighbors, iteratively updates their states until convergence using GNN smoothing properties (Li et al., 2018), and ensures persistent influence by adding augmented future neighbors. T-Spear (Lee et al., 2024) introduced a model-agnostic poisoning attack for continuous-time dynamic graphs that corrupts training data while maintaining realistic temporal patterns. It uses a surrogate model to identify candidate edges for perturbation, enforces constraints on temporal distribution and node connectivity, and samples adversarial edge features using Kernel Density Estimation. While T-Spear relies on heuristic constraints, our approach optimizes an objective function for adversarial node pair selection, though we adopt similar temporal perturbation strategies using Gaussian priors for modeling time differences between edges.

## 3 PROBLEM STATEMENT

In this section, we formulate the problem of adversarial attacks on temporal graphs. We start for introducing TG notations.

**Definition 3.1** (Continuous-Time Dynamic Graph). A Continuous-Time Dynamic Graph (CTDG) $\mathcal{G}$ is defined as a tuple $\mathcal{G} = (\mathcal{V}, \mathcal{E}, \mathcal{T}, \mathcal{F})$, where:

- $\mathcal{V}$ is the set of vertices

- $\mathcal{E} \subseteq \mathcal{V} \times \mathcal{V} \times \mathbb{R}_{\geq 0}$ is the set of timestamped edges: $(u, v, t) \in \mathcal{E}$ means edge $(u, v)$ is observed at time $t$.

- $\mathcal{T} = \{t_1, \ldots, t_{|\mathcal{E}|}\} = \{t \mid (u, v, t) \in \mathcal{E}\}$ is the set of timestamps with $0 \leq t_1 \leq \ldots \leq t_{|\mathcal{E}|}$

- $\mathcal{F} = \{\mathbf{f}_1, \ldots, \mathbf{f}_{|\mathcal{E}|}\} \subset \mathbb{R}^D$ is the set of edge features (optional)

**Definition 3.2** (Temporal Link Prediction). Given a CTDG $\mathcal{G}$, temporal link prediction involves learning a function $h_\theta : \mathcal{V} \times \mathcal{V} \times \mathbb{R} \to [0, 1]$ that estimates the probability of an edge existing between nodes $u, v \in \mathcal{V}$ at time $t \in \mathbb{R}$. The function $h_\theta$ can utilize all information in $\mathcal{G}$ up to time $t$ to make its prediction.

Dynamic graphs can be perturbed in multiple ways: by modifying the graph structure (structural perturbation), edge features (contextual perturbation), or edge timestamps (temporal perturbation). Moreover, the perturbations may be additive or modify elements in the clean graph. Formally, we define an adversarial attack as:

**Definition 3.3** (CTDG Adversarial Attacks). Let $\mathcal{G}' \in \Phi(\mathcal{G})$ be the perturbed graph chosen from the set of permissible perturbations $\Phi(\mathcal{G})$ in the vicinity of the clean graph $\mathcal{G}$. Then, an adversarial attack is concerned with the following optimization problem:

$$\max_{\mathcal{G}' \in \Phi(\mathcal{G})} \mathcal{L}_{attack}(h_\theta(\mathcal{G}'), \mathcal{G}) \tag{1}$$

where $\mathcal{L}_{attack}$ is a loss function that quantifies the model's prediction error on the perturbed graph $\mathcal{G}'$ vs. the clean graph $\mathcal{G}$. The attacker aims to minimize the model's link prediction performance by optimizing Equation 1 where $L_{\text{attack}}$ is the loss function on the perturbed graph $G'$ in relation to model $h_\theta$. For link prediction, this is typically the negative Mean Reciprocal Rank (MRR) or margin-based loss designed to degrade ranking performance.

**TGNN Memory Modules:** For TGNNs with memory modules (e.g., TGN, TNCN), each node $v$ maintains a memory state $\mathbf{m}_v^{(t)}$ that evolves as edges arrive sequentially. At each inference step when processing edges $E_t = \{(u_i, v_i, t_i, f_i)\}_{i=1}^{|E_t|}$, the model updates memory as:

$$\mathbf{m}_v^{(t+1)} = \text{UpdateMemory}_\theta(\mathbf{m}_v^{(t)}, E_t) \tag{2}$$

where the memory state is then used to generate predictions: $\hat{\mathbf{y}} = h_\theta(\mathbf{m}^{(t+1)}, E_t)$. The key insight is that adversarial edges corrupt memory: $\mathbf{m}_v^{(t+1)} = \text{UpdateMemory}_\theta(\mathbf{m}_v^{(t)}, E_t \cup E_{\text{adv}})$, and this corrupted state persists across all future time steps, compounding the attack's impact. This memory coupling distinguishes temporal attacks from static graph attacks where perturbations are localized. Figure Figure 1 visualizes this memory pollution propagation across batches.

**Threat Model and Attack Setting.** To establish clear baselines for comparing adversarial attacks on TGNNs, we formally specify our threat model and attacker capabilities. Understanding these assumptions is critical for practitioners deploying TGNNs and for researchers developing defense mechanisms. We consider a white-box evasion attack in the test-time setting, where attacks occur after model deployment without access to training data. This setting is appropriate for scenarios where adversaries target deployed systems, such as fraud detection or intrusion detection systems that have already been trained and fixed. We assume the attacker has *full white-box access* to the victim TGNN, including: (1) model architecture and hyperparameters, (2) all trainable parameters (e.g., embedding matrices, attention weights), (3) node and edge embeddings at test time, and (4) model gradients with respect to the loss function. This white-box formulation establishes an *upper bound* on attack effectiveness. This threat model aligns with established practices in adversarial robustness literature Xu et al. (2019); Geisler et al. (2021); Günnemann (2022) for two key reasons. First, white-box attacks establish necessary conditions for vulnerability, if a model resists white-box attacks, it is inherently more robust to restricted black-box or transfer attacks. Second, white-box attacks model realistic scenarios where adversaries have significant reconnaissance capabilities (e.g., insider threats, model extraction attacks, or organizations with shared infrastructure).

**Attack Constraints and (Un-) Noticeability.** In the seminal work on adversarial attacks on deep learning methods, Szegedy et al. (2014) proposed the concept of unnoticeability ("imperceptibly tiny perturbations") since it usually does not alter the true semantics ("underlying class" in their classification setting). Hence, a key requirement for adversarial attacks is that its perturbations should be unnoticeable. This is especially true if we cannot rely on application-specific insights about the true semantics (e.g., see Geisler et al. (2022)). Following the best practices of adversarial robustness in the graph domain and beyond (Günnemann, 2022), the attacker can perform the following operations within a fixed perturbation budget: (1) *structural perturbations*: insert new edges $(u, v, t)$ among existing nodes by modifying the edge set $E$; (2) *temporal perturbations*: choose timestamps $t$

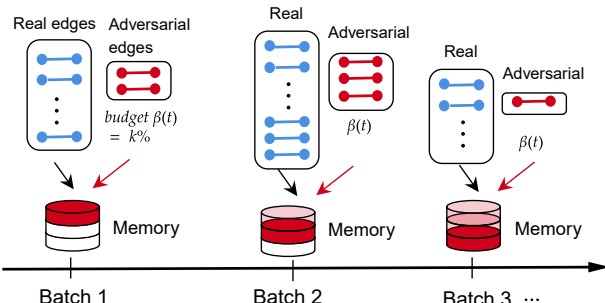

Figure 2: The adversarial edges are inserted at each batch as a small portion of the real edges (constrained by budge $\beta(t)$). The attacks gradually corrupt the victim model's memory step by step.

---

**Algorithm 1:** TR-BCD Evasion Attack on TGNN

---

**Input:** Test graph $\mathcal{G}_{\text{test}}$, TGNN $h_\theta$ with memory $M$
**Params.:** Attack budget $\beta(t)$
1 **for** $\mathcal{E}, \mathcal{T}, \mathcal{F}$ *in* $\text{batch}(\mathcal{G}_{\text{test}})$ **do**
2      $\tilde{\mathcal{E}}, \tilde{\mathcal{T}}, \tilde{\mathcal{F}} \leftarrow \text{TR-BCD}_{\text{step}}(\mathcal{E}, \mathcal{T}, \mathcal{F}, h_\theta, M, \beta(t))$ ;                `// See Algorithm 2`
3      $M \leftarrow \text{UpdateMemory}_{h_\theta}(M, \tilde{\mathcal{E}}, \tilde{\mathcal{T}}, \tilde{\mathcal{F}})$;
4      $M, \hat{y} \leftarrow h_\theta(M, \mathcal{E}, \mathcal{T}, \mathcal{F})$;
5 **end**
6 **return** all predictions $\hat{y}$ ;

---

for adversarial edges such that perturbations reflect realistic temporal patterns; and (3) *contextual perturbations*: modify edge feature vectors $\mathbf{f}$ within a bounded Euclidean distance (for attributed graphs). All perturbations are additive: the attacker adds adversarial edges to the clean graph without removing or modifying existing edges. The total number of inserted edges is constrained by budget $\delta_t$, defined as a percentage of edges in each test batch.

To ensure attacks remain realistic and unnoticeable, we enforce three key constraints. *Temporal Causality:* All adversarial edge timestamps must respect the temporal history of affected nodes. Specifically, for any adversarial edge $(u, v, t_{\text{adv}})$, we require $t_{\text{adv}} \geq t_{\min}(u, v)$, where $t_{\min}(u, v)$ is the earliest observed interaction time for nodes $u$ and $v$ in the clean graph. This prevents causally inconsistent attacks. *Budget Constraint:* The number of edges inserted in each batch cannot exceed $\lfloor \delta_t \cdot |E_{\text{batch}}| \rfloor$, where $|E_{\text{batch}}|$ is the number of benign edges in the current batch. *Feature Deviation Bound:* For contextual perturbations, edge feature modifications are bounded by $\|\mathbf{f}_{\text{adv}} - \mathbf{f}\|_2 \leq \epsilon_f$ for a specified threshold $\epsilon_f$.

## 4 TR-BCD ATTACK

We propose Temporally-aware Randomized Block Coordinate Descent (*TR-BCD*), a greedy gradient-based discrete optimization method for adversarial evasion attacks in CTDGs. To alleviate the prohibitive memory requirements of the optimization problem in Equation (1) with the challenging additive perturbations, we follow two strategies: (1) we greedily apply the attack before each benign batch at inference time; (2) we leverage Randomized Block Coordinate Descent.

**Greedily over time.** As we detail in Algorithm 1, we greedily optimize for the adversarial perturbations. That is, in each time step, we choose adversarial perturbations given the information up to this point in time. Thereafter, we use the perturbed edges to update the model's memory. This greedy procedure reduces the memory complexity from $\Theta(|\mathcal{E}||\mathcal{V}|^2)$ to $\Theta(|\mathcal{V}|^2)$. However, due to the perturbed memory, we then indirectly affect the model's predictions. We refer to Figure 2 for a graphical illustration of the memory pollution process.

**Modeling of Edge Insertions.** In each call of TR-BCD, we choose up to $\beta(t)$ edges to insert. We further simplify the procedure by allowing solely a distinct set of edges in each adversarial memory update (line 3 in Algorithm 1). Thus, we can model the possible insertions using a matrix

---
**Algorithm 2:** Step of Temporally-Aware Randomized Block Coordinate Descent (TR-BCD)

---
**Input:** $\mathcal{E}, \mathcal{T}, \mathcal{F}$, TGNN $h_\theta$, Memory $M$, Budget $\beta(t)$
**Params.:** Block size $b$, $\mathcal{L}_{attack}$, time std. $\sigma_{\Delta t}$, feat. budget $\epsilon$
**Initialize:** $\mathbf{B} \in [0,1]^{|\mathcal{V}| \times |\mathcal{V}|}$ and sample block of size $b$.
// Temporal Perturbation:
1 $\tilde{\mathcal{T}} \leftarrow \min(\mathcal{T}) + \Delta t$, where $\Delta t \in \mathbb{R}^b_{\geq 0}$ s.t. $\Delta t \sim \mathcal{N}(0, \sigma^2_{\Delta t})$;
// Structural Perturbation:
2 $\tilde{\mathcal{F}} \leftarrow$ Sample from edges where $t < \min(\mathcal{T})$;
3 $\tilde{M} \leftarrow \text{UpdateMemory}_{h_\theta}(M, \tilde{\mathcal{E}}(\mathbf{B}), \tilde{\mathcal{T}}, \tilde{\mathcal{F}})$;
4 $\hat{\mathbf{y}} \leftarrow h_\theta(\tilde{M}, \tilde{\mathcal{E}}(\mathbf{B}), \tilde{\mathcal{T}}, \tilde{\mathcal{F}})$;
5 $\tilde{\mathcal{E}}, \tilde{\mathcal{T}}, \tilde{\mathcal{F}} \leftarrow$ get top $\beta$ of $\tilde{\mathcal{E}}, \tilde{\mathcal{T}}, \tilde{\mathcal{F}}$ according to $\nabla_{\mathbf{B}} \mathcal{L}_{attack}(\hat{\mathbf{y}})$;
// Optional Contextual Perturbation:
6 $\tilde{\mathcal{F}} \leftarrow \tilde{\mathcal{F}} + \epsilon \cdot \text{sign}(\nabla_{\tilde{\mathcal{F}}} \mathcal{L}_{attack}(\hat{\mathbf{y}}))$;
7 **return** $\tilde{\mathcal{E}}, \tilde{\mathcal{T}}, \tilde{\mathcal{F}}$;

---

$\mathbf{B} \in \{0,1\}^{|\mathcal{V}| \times |\mathcal{V}|}$, where a one denotes an edge insertion–much alike an adjacency matrix of a static (non-multi) graph. The optimization problem for each time step sets up to $\beta(t)$ entries/edges in $\mathbf{B}$ to one with the goal of maximizing the loss $\mathcal{L}_{attack}$.

**Gradient-Based Procedure to Choose Discrete Edges.** To use a gradient based attack on discrete edges, we follow Xu et al. (2019) and relax the edges from $\{0,1\}^{|\mathcal{V}| \times |\mathcal{V}|}$ to $[0,1]^{|\mathcal{V}| \times |\mathcal{V}|}$ during the attack. In other words, we introduce *edge weights*. We can then use the gradient $\nabla_{\mathbf{B}} \mathcal{L}_{attack}$ to optimize over these weights. While we study the greedy perturbations based on the gradient $\nabla_{\mathbf{B}} \mathcal{L}_{attack}$, similar to GR-BCD of Geisler et al. (2021), it would also be possible to use other optimization procedures (Geisler et al., 2021; Gosch et al., 2023).

**Randomized Block Coordinate Descent.** Naïvely optimizing over $\mathbf{B}$, e.g., using gradient descent, would require keeping $\Theta(|\mathcal{V}|^2)$ parameters in memory and incur a cost of $\Theta(|\mathcal{V}|^2)$ for gradient evaluation. For this reason, we use Randomized Block Coordinate Descent instead. Coordinate Descent refers to optimizing over a single parameter dimension in each step, Block Coordinate Descent extends this to multiple dimensions, and Randomized Block Coordinate Descent chooses the dimensions to be optimized in each step randomly. Due to its efficiency, RBCD is an established method for large-scale gradient-based optimization (Wright, 2015).

**Gradient-Based Edge Selection:** The gradient-based edge selection procedure can be formally expressed as:

$$\text{Step 1 (Relaxation)}: \quad B \in [0,1]^{|V| \times |V|} \tag{3}$$

$$\text{Step 2 (Gradient)}: \quad \nabla_B \mathcal{L}_{\text{attack}}(\hat{\mathbf{y}}) \tag{4}$$

$$\text{Step 3 (Selection)}: \quad B^* = \text{top-}\beta(t)\{B[i,j] : \nabla_B \mathcal{L}_{\text{attack}}[i,j] \text{ is largest}\} \tag{5}$$

where $B[i,j] = 1$ indicates insertion of adversarial edge $(i,j)$, and we greedily select the $\beta(t)$ entries with highest gradient magnitude. This relaxation from discrete $\{0,1\}$ to continuous $[0,1]$ enables gradient flow during optimization, following standard practice in adversarial attacks on graphs (Xu et al., 2019). In our context and due to the greedy flipping of entries in $\mathbf{B}$, we sample $b$ dimensions and only need to evaluate the gradient towards the current batch. This reduces the memory complexity from $\Theta(|\mathcal{V}|^2)$ to $\Theta(b)$. We can greedily flip $\beta(t)$ entries in $\mathbf{B}$ in a single iteration if $\beta(t) \leq b$.

**Candidate Sampling.** We find that naïvely sampling from all possible $\mathcal{V} \times \mathcal{V}$ edges can yield suboptimal results since we usually choose $b \ll |\mathcal{V}|^2$ for its lower computational cost. To keep the computational footprint low, we remedy this limitation via a specialized candidate sampling strategy for TGNNs. Specifically, we optionally over-represent the so-called negative historical edges of the current batch. We call our method with fully random initial sampling *TR-BCD-random* and with over-represented negative historical edges *TR-BCD-mixed*. For *TR-BCD-mixed*, we randomly sample 50% of the block from all possible $\mathcal{V} \times \mathcal{V}$ edges and the other 50% from the historical negative edges of the current nodes. This mixed sampling strategy exploits the fact that TGNNs are trained with historical negatives as challenging hard negatives edges that appeared historically but are absent at the

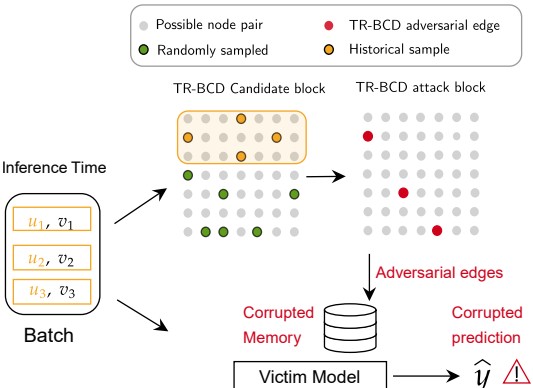

Figure 3: Candidate selection in TR-BCD. For each batch, TR-BCD first constructs a candidate pool based on *random* (green) and *historical* (orange) candidate edges for the adversarial attack.

query time. By over-representing these edges (50% of the sampled block), we bias the gradient-based optimization toward the regions of the loss landscape where TGNNs are most sensitive. Empirically, this recency-biased approach achieves 7.6% improvement over random sampling (50.78% vs. 43.16% MRR drop on Wikipedia, see Table 1), validating this architectural insight. We illustrate the candidate sampling scheme in Figure 3.

**Computational Complexity.** The computational space and time complexity of a single pass of Algorithm 2 is $\mathcal{O}(b)$, assuming that the TGNN's forward pass, backward pass, and memory update are also linear in $b$. Hence, the complexity is $\mathcal{O}(|\mathcal{E}_{\text{test}}|b) = \mathcal{O}(|\mathcal{E}_{\text{test}}|)$ for the entire test set (Algorithm 1), assuming a constant batch size and $b \ll |\mathcal{E}_{\text{test}}|$.

## 5 EXPERIMENTS

In this section, we evaluate the robustness of TGNNs under evasion attacks and demonstrate the effectiveness of our TR-BCD method. For empirical experiments, we use six widely used datasets for link prediction in CTDGs (Poursafaei et al., 2022). We choose a mix of bipartite and non-bipartite, attributed and non-attributed datasets to study the effect of adversarial attacks across different types of graphs. Dataset details and statistics are reported in Appendix B. We report detailed sensitivity analysis of TR-BCD on the effect of contextual perturbation budget and block size in Appendix E. We use the widely-used evaluation procedure from (Huang et al., 2024) where the MRR ranking metric is used to evaluate link prediction to predict the true destination from multiple negative edges (including random and historical negatives). In our evaluation, we use all possible negative edges for the Wikipedia dataset, and 100 negative edges per positive edge for the remaining datasets. we compare the TGNN performance with no perturbation and with perturbation on the test set. To evaluate the effect of adversarial attacks, we select strong TGNN models including TGN and TNCN as the base model to inject attack with, referred to as the *victim models*. Both models have a memory module that records past node interactions and performs test-time memory updates. The adversarial attacks are injected into the model memory (see line 3 Algo 2). Victim model training details are in Appendix C. We repeat each experiment 5 times and report the metrics mean and standard deviation.

**Baselines.** we include two heuristic baselines for structural perturbation: *random* and *historical* baseline. The *random baseline* generates each of the adversarial edges independently and randomly from the space of all possible negative node pairs. The *historical baseline* considers historical negative edges as defined in (Poursafaei et al., 2022), meaning edges that were observed before but were not present at the current time. These negative edges are challenging for TGNN models as they were encountered previously but currently non-existing. Lastly, we compare with MemStranding (Dai et al., 2023), a sophisticated evasion attack designed for TGNNs with memory modules. Unlike our TR-BCD approach that distributes adversarial edges across temporal batches, MemStranding operates as a single-shot evasion attack that inserts a burst of fake edges at a single timestamp to corrupt node memory states.

Table 1: Structural perturbation attack results for dynamic link property prediction on CTDG datasets under a 5% perturbation budget (i.e., 5% of the test edges are perturbed). We compare two variants of TR-BCD: one with fully random initial sampling (*TR-BCD-random*) and another with a mixed strategy using 50% initialization based on historical negative edges (*TR-BCD-mixed*). We include two generally-applicable baselines: random and Historical attack (insertion of historical negatives). Additionally, we evaluate *MemStranding attack* Dai et al. (2023), which targets 5% of nodes as victims by strategically inserting fake neighbors to disrupt temporal graph dynamics. N/A means not applicable due to the lack of edge attributes and we mark noticeable/non-evasive attacks in grey. Performance is reported in Mean Reciprocal Rank (MRR) and averaged over 5 trials.

| Model | Attack | Wikipedia | Reddit | Lastfm | Enron | UCI | MOOC |
|---|---|---|---|---|---|---|---|
| **TGN** | No Attack | $0.3929$ ±0.0366 | $0.4550$ ±0.0485 | $0.1370$ ±0.0172 | $0.2662$ ±0.0167 | $0.3058$ ±0.0104 | $0.1574$ ±0.0590 |
| | Random Attack | $0.3752$ ±0.0254 | $0.4299$ ±0.0404 | $0.1310$ ±0.0537 | $0.2625$ ±0.0128 | $0.2939$ ±0.0119 | $0.1476$ ±0.0479 |
| | Historical Attack | $0.3541$ ±0.0351 | $0.4436$ ±0.0466 | $0.1207$ ±0.0468 | $0.2417$ ±0.0245 | $0.3194$ ±0.0119 | $0.1330$ ±0.0381 |
| | Memstranding | $0.3287$ ±0.0188 | $0.4220$ ±0.0502 | N/A | N/A | N/A | $0.0884$ ±0.0267 |
| | TR-BCD-random (ours) | $0.2233$ ±0.0546 | **$0.3350$** ±0.1191 | $0.1306$ ±0.0497 | $0.2558$ ±0.0088 | $0.2893$ ±0.0136 | $0.1245$ ±0.0491 |
| | TR-BCD-mixed (ours) | **$0.1934$** ±0.0587 | $0.3689$ ±0.0980 | **$0.1080$** ±0.0421 | **$0.2321$** ±0.0229 | **$0.2857$** ±0.0181 | $0.1351$ ±0.0367 |
| | Max perf. drop | -50.78% | -26.38% | -21.17% | -12.81% | -6.57% | -20.90% |
| **TNCN** | No Attack | $0.7207$ ±0.0009 | $0.7228$ ±0.0064 | $0.3632$ ±0.0029 | $0.4257$ ±0.0123 | $0.4839$ ±0.0030 | $0.2521$ ±0.0192 |
| | Random Attack | $0.7197$ ±0.0015 | $0.7224$ ±0.0055 | $0.3591$ ±0.0031 | $0.4308$ ±0.0155 | $0.4795$ ±0.0035 | $0.2187$ ±0.0224 |
| | Historical Attack | $0.7167$ ±0.0015 | $0.7204$ ±0.0059 | $0.3564$ ±0.0031 | $0.3999$ ±0.0171 | $0.4806$ ±0.0055 | $0.2225$ ±0.0182 |
| | Memstranding | $0.7068$ ±0.0015 | $0.7196$ ±0.0096 | N/A | N/A | N/A | $0.1903$ ±0.0138 |
| | TR-BCD-random (ours) | $0.7057$ ±0.0167 | **$0.5869$** ±0.1688 | $0.3410$ ±0.0080 | $0.4258$ ±0.0172 | **$0.4793$** ±0.0025 | **$0.1164$** ±0.0236 |
| | TR-BCD-mixed (ours) | **$0.7021$** ±0.0137 | $0.6681$ ±0.0397 | $0.3557$ ±0.0032 | **$0.3980$** ±0.0232 | $0.4836$ ±0.0047 | $0.1258$ ±0.0200 |
| | Max perf. drop | -2.58% | -18.80% | -6.11% | -6.51% | -0.95% | -53.83% |

Table 2: Comparison of vanilla TR-BCD (using random initialization with structural perturbation) and TR-BCD augmented with contextual perturbation via FGSM ($\epsilon = 0.3$). Performance is measured in Mean Reciprocal Rank (MRR) and averaged over 5 trials.

| Model | Attack | Wikipedia | Reddit | MOOC |
|---|---|---|---|---|
| **TGN** | No Attack | $0.3929$ ±0.0366 | $0.4550$ ±0.0485 | $0.1574$ ±0.0590 |
| | TR-BCD | $0.2233$ ±0.0546 | **$0.3350$** ±0.1191 | **$0.1245$** ±0.0491 |
| | TR-BCD (FGSM) | **$0.2073$** ±0.0597 | $0.3403$ ±0.1063 | $0.1273$ ±0.0423 |
| **TNCN** | No Attack | $0.7207$ ±0.0009 | $0.7228$ ±0.0064 | $0.2521$ ±0.0192 |
| | TR-BCD | $0.7057$ ±0.0167 | $0.5869$ ±0.1688 | $0.1164$ ±0.0236 |
| | TR-BCD (FGSM) | **$0.7043$** ±0.0097 | **$0.5807$** ±0.1802 | **$0.1149$** ±0.0297 |

**Structural Perturbation Results.** First, we examine how robust are TGNNs to structural adversarial attacks. In Table 1, we report MRR performance on link prediction for the two victim models: TGN (Rossi et al., 2020) and TNCN (Zhang et al., 2024) across the considered datasets with and without perturbations. The attack budget $\beta(t)$ is set to be $5\%$ as it is an unnoticable amount (see Section 3 for the discussion on unnoticability). As shown in Table 1, the victim models are highly vulnerable to TR-BCD's attack across all datasets, with up to 53.83% drop in MRR for the MOOC dataset. Note that because Memstranding Dai et al. (2023) requires edge features to perturb on, it is *not applicable* (N/A) for Enron and UCI datasets. In comparison, TR-BCD applies to all datasets and not restricted by attributes. While both random and historical attack baselines can cause a small performance drop from the victim model, choosing the adversarial edge based only on heuristics is suboptimal. This is most evident in the Wikipedia and Lastfm datasets where the adversarial edges picked by TR-BCD are significantly more effective than both baselines (with up to $0.16$ difference in MRR drop in Wikipedia). Therefore, TGNN models are highly susceptible to gradient-based attacks. Random and historical baselines achieve mostly similar performance across all datasets. However, using both random and historical negative edges as candidates to sample has proven to be an effective variant of TR-BCD (namely, TR-BCD-mixed). The intuition is that TR-BCD can learn to select strong adversarial samples from both categories based on the victim model's gradient. Interestingly, the dataset with most performance drop is distinct between the two victim models. Particularly, TGN has a 50.78% drop in MRR on the Wikipedia dataset, while TNCN has a 53.83% drop in MRR for the MOOC dataset. This shows that TGNNs might be vulnerable to attacks at different network domains thus highlighting the importance of benchmarking their robustness in a wide range of networks.

**The Effect of Attack Budget.** Here, we investigate the effect of varying the attack budget $\beta(t)$ on the performance of the victim model. Figure 4 shows the performance of the victim models under

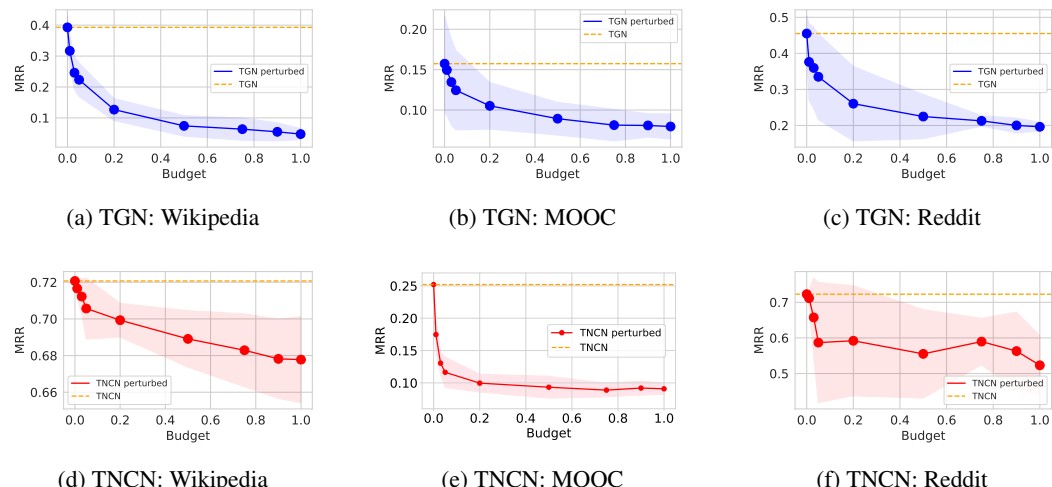

Figure 4: TGN (top) and TNCN (bottom) link prediction performance under TR-BCD structural perturbation with varying budget for Wikipedia (left), Reddit (center), and MOOC (right) datasets.

different budgets for TR-BCD attacks on three datasets. The orange line in each plot represents the performance of the unperturbed base model, while the other line plots depict the corresponding drop in MRR as the attack budget increases. These plots show how sensitive each model is to the intensity of adversarial perturbations. For both TGN and TNCN, model performance degrades rapidly with attack budgets just under 20%, demonstrating that even a relatively small number of adversarial edges can have a severe impact. This finding indicates an ideal trade-off region where a modest attack budget is sufficient to induce significant performance drops without requiring many perturbations. Notably, on the Wikipedia and MOOC datasets, both TGN and TNCN are exhibiting an exponential decay in MRR as the attack budget increases. Furthermore, beyond a certain budget threshold, we observe a plateau in performance degradation where additional adversarial edges produce diminishing impacts. This plateau might be due to the fact that model weights are frozen during attacks, thus retaining their learned knowledge despite the memory being corrupted. These observations underscores the important of studying adversarial robustness of TGNNs.

**Contextual Perturbations.** Here, we evaluate the effectiveness of adding contextual perturbations in TR-BCD. As noted in Algorithm 2 (Step 6), we can optionally apply contextual perturbations, e.g., using the Fast Gradient Sign Method (FGSM) (Goodfellow, 2014) on the edge features for graphs that include edge attributes. FGSM aims to maximize the loss of a neural network by modifying the input data in the direction that increases the model's error, thereby probing the model's sensitivity to changes in its feature space. In Table 2, we compare the results of TR-BCD with and without contextual perturbations on the edge features. We report results only for datasets containing edge features (see Appendix B). Overall, our experiments indicate that adding contextual perturbations yields little MRR drop compared to using solely structural perturbations. With the exception of the Wikipedia dataset, where the TGN suffers an additional 2% performance drop due to the added contextual perturbation, results suggest that TGNNs are primarily vulnerable to structural attacks.

**Evasiveness and Anomaly Detection.** An important measure of adversarial attacks is their *evasiveness*: the ability of perturbations to remain undetected by security systems while maintaining their adversarial effectiveness. Evasive attacks should be as close to normal behavior as possible to avoid triggering anomaly detection mechanisms. To evaluate the evasiveness of attacks, we employed SPOTLIGHT (Eswaran et al., 2018), a strong anomaly detection algorithm designed for streaming graphs. The algorithm's effectiveness stems from its ability to detect sudden appearances or disappearances of dense subgraphs, checking against the evasiveness of an attack. Figure 5 shows that TR-BCD demonstrates superior evasiveness compared to single-shot attacks like Memstranding. TR-BCD maintains relatively stable anomaly scores throughout the attack period, with only modest increases that remain within the normal range of variation. This is because TR-BCD attacks are designed to evasive, only inserting a small number of edges per batch. In contrast, the MemStranding attack, being a single-shot approach, introduces a sudden burst of adversarial edges at a specific time

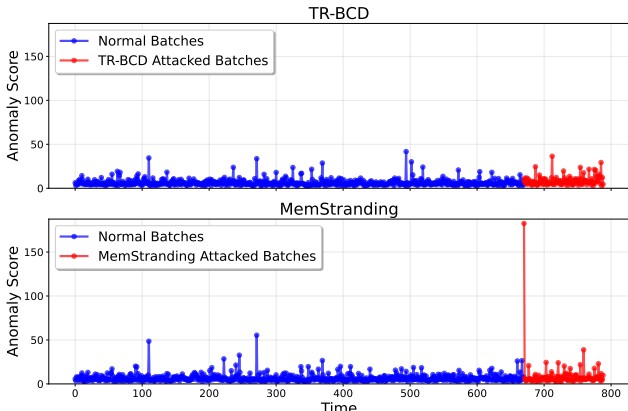

Figure 5: Comparison of SPOTLIGHT anomaly scores over time for TGN on Wikipedia dataset under different attack scenarios. Normal batches are shown in blue, while *TR-BCD attack* (top) and *MemStranding attack* (bottom) batches are highlighted as red stars to demonstrate their distinct impact on anomaly detection scores.

Table 3: Transfer attack results: Link prediction MRR ($\pm$ SD, % drop) for same-model and cross-model (transfer) attacks on TGN and TNCN.

| Model | Attack | Wiki | Reddit | Mooc |
|-------|--------|------|--------|------|
| **TGN** | No attack | 0.4105 $\pm_{0.0195}$ | 0.4719 $\pm_{0.0471}$ | 0.1428 $\pm_{0.0503}$ |
| | Attack with same model | **0.2533** $\pm_{0.0674}(-38.29\%)$ | **0.3433** $\pm_{0.1499}(-27.25\%)$ | **0.1314** $\pm_{0.0241}(-7.98\%)$ |
| | Transfer attack with TNCN | 0.2890 $\pm_{0.0317}(-29.60\%)$ | 0.3688 $\pm_{0.1316}(-21.85\%)$ | 0.1327 $\pm_{0.0369}(-7.07\%)$ |
| **TNCN** | No attack | 0.7212 $\pm_{0.0009}$ | 0.7264 $\pm_{0.0014}$ | 0.2439 $\pm_{0.0154}$ |
| | Attack with same model | **0.7122** $\pm_{0.0028}(-1.25\%)$ | **0.6661** $\pm_{0.0764}(-8.30\%)$ | **0.1149** $\pm_{0.0118}(-52.89\%)$ |
| | Transfer attack with TGN | 0.7126 $\pm_{0.0016}(-1.19\%)$ | 0.6803 $\pm_{0.0330}(-6.35\%)$ | 0.2109 $\pm_{0.0127}(-13.53\%)$ |

point. The sharp increase in anomaly scores clearly indicates the presence of anomalous activity, demonstrating its disadvantage of being easily detected by an anomaly detection algorithm.

**Cross-Model Transfer Attacks**. Transfer attacks investigate the adversarial vulnerability of a victim model by crafting perturbations using a different, pretrained attacker model. This approach tests whether the adversarial edge or feature perturbations optimized for one TGNN architecture are also effective against another. It helps in understanding shared weaknesses in model families. In our experiments, we evaluate transfer attacks by attacking TGN with a pretrained TNCN, and conversely, attacking TNCN with a pretrained TGN. Table 3 shows that attacks trained on TGN transfer to TNCN and vice-versa with 25-95 % of same-model performance, suggesting temporal vulnerabilities generalize across architectures. Notably, transfer attacks on TNCN with TGN are much weaker for the MOOC dataset (-13.5%) compared to direct attacks (-52.9%), showing that transferability is influenced by both model and dataset characteristics. Transfer attacks can thus induce substantial performance drop in both TGN and TNCN victims, but are slightly less effective than attacks tuned for the same model. This suggests that while adversarial perturbations generalize across model families to an extent, there remain architecture-specific vulnerabilities.

## 6 CONCLUSION

In this work, we conducted a comprehensive study of adversarial robustness in Temporal Graph Neural Networks (TGNNs) operating on Continuous-Time Dynamic Graphs (CTDGs). We identified that TGNNs can be highly vulnerable to adversarial attacks with up to 53% drop in performance. Our investigation spanned diverse real-world datasets, including both bipartite and non-bipartite graphs, with and without edge features. Notably, our experiments revealed that structural perturbations have a more substantial impact compared to contextual feature perturbations, suggesting TGNNs are highly vulnerable to attacks on the temporal graph topology. We hope this work serve as foundation for future studies aiming at studying adversarial robustness.

## REPRODUCIBILITY STATEMENT

We provide an anonymized code repository at https://anonymous.4open.science/r/temporal-adversarial-02B3, which contains the implementation of our model and experimental setup to ensure reproducibility. Dataset details and access links can be found in Appendix B. Experimental details are recorded in Appendix C.

## ETHICS STATEMENT

In this work, we examine the robustness of Temporal GNNs to adversarial attacks and proposed a novel adversarial attack TR-BCD for this purpose. It is possible that the studied adversarial attack or similar attacks might be considered by ill-intentioned third party and the goal of this paper is to warn ML practitioners of such risks. Overall, we are convinced that the benefits outweigh the risks. Document and open-source the adversarial attack study will help enable researchers to design more robust models against such attack. We firmly believe that open research into such vulnerabilities of models allows researchers and practitioners to identify the problems and address them with strong defences. Moreover, due to our setting being a white-box setting, our attack is less directly applicable for real-world malicious actors.

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

## A    LLM USAGE

We acknowledge the use of LLMs to assist in improving the writing of this paper. All content, ideas, and results are our own. The LLM helped improve clarity, grammar, style, and LaTeX formatting.

Table 4: Statistics of the datasets used in our experiments

| Dataset | Domain | # Nodes | # Edges | # Unique Edges | # Edge features | Bipartite | Duration |
|---------|--------|---------|---------|----------------|-----------------|-----------|----------|
| Wikipedia | Social | 9,227 | 157,474 | 18,257 | 172 | ✓ | 1 month |
| Reddit | Social | 10,984 | 672,447 | 78,516 | 172 | ✓ | 1 month |
| MOOC | Interaction | 7,144 | 411,749 | 178,443 | 4 | ✓ | 17 months |
| LastFM | Interaction | 1,980 | 1,293,103 | 154,993 | - | ✓ | 3 years |
| Enron | Social | 184 | 125,235 | 3,125 | - | ✗ | 8 months |
| UCI | Social | 1,899 | 59,835 | 20,296 | - | ✗ | 196 days |

## B  DATASET DETAILS

The statistics of these datasets are listed in Table 4. The details of each dataset is as follows:

- Wikipedia (Kumar et al., 2019) enlists edits on Wikipedia pages over one month. It is a bipartite graph with edges between users and wiki pages which are modeled as nodes. Each edge carries a 172-dimensional vector representing the page edits.
- Reddit (Kumar et al., 2019) stores user posts on subreddits over one month. It is a bipartite graph with 172-dimensional edges between users and subreddits.
- MOOC (Kumar et al., 2019) models the interaction of users with online course content spanning over 17 months. It is a bipartite graph with the edges representing interaction of a user with one of 97 course units. The edges have 4 features.
- LastFM (Kumar et al., 2019) is a bipartite graph featuring user-to-song relations where each edge representing whether one of the 1000 users listened to one of the 1000 most listened songs over a period of one month. The dataset has no edge features.
- UCI (Panzarasa et al., 2009) contains interactions on an online social network between students of University of California, Irvine. It is a non-attributed, non-bipartite graph.
- Enron (Shetty & Adibi, 2004) stores information about email exchanges between employees of ENRON energy over 3 years. The dataset is non-bipartite and has no edge features.

These datasets can be accessed from (Poursafaei et al., 2022) via the link `https://zenodo.org/records/7213796#.Y8QicOzMJB2`.

## C  EXPERIMENT DETAILS

**Evaluation setting.** Recent work showed that link prediction on temporal graphs requires challenging negative samples (Poursafaei et al., 2022) and ranking metrics for robust evaluation. Therefore, we use the same evaluation procedure as in (Huang et al., 2024) where the link prediction task is treated as a ranking problem and multiple negative samples per positive edge are used to compute the Mean Reciprocal Rank (MRR). These negative edges are a mix of *random* and *historical* negative edges. Historical negative edges are edges that were observed in previous timestamps but not observed currently thus being challenging for models that rely on memorization (Poursafaei et al., 2022). In our evaluation, we use all possible (999) negative edges for the Wikipedia dataset, and 100 negative edges per positive for the remaining datasets. The MRR metric takes its values in $(0, 1]$ and computes the reciprocal rank of the true edge among the negative edges. To understand the robustness of TGNN models, we examine their MRR performance under no perturbation and with perturbation on the entire test set.

**Victim Models.** We examine state-of-the-art TGNN models including TGN (Rossi et al., 2020) and TNCN (Zhang et al., 2024) for evaluating robustness to adversarial attacks, we refer to them as the *victim models*. Both models have a memory module that records past node interactions and performs test-time updates (i.e. the memory is updated by test-time data). The adversarial attacks are injected into the model memory (see lines $7 - 8$ Algo 1).

We train both the models in batches of 200 edges using the Adam (Kingma, 2014) optimizer with learning rate=$1e - 4$. The dataset is chronologically split into a train-val-test split of edges in the $0.75 - 0.15 - 0.15$ proportion. The input edge features are normalized to have zero mean and unit variance. We train for a maximum of 50 epochs using early stopping with a patience of 5 epochs. We repeat each experiment 5 times and report the metrics mean and standard deviation over all the runs.

**Baselines.** For our experiments, we consider two heuristic baselines for structural perturbation: *random* and *historical* baseline. The *random baseline* generates each of the adversarial edges independently and randomly from the space of all possible negative node pairs. The source and destination nodes are picked randomly from the set of source and destination ids in the dataset. The timestamp is uniformly sampled between the minimum and maximum values of timestamps present in the dataset. For attributed graphs, the adversarial edge features are generated by sampling from a normal distribution. The edges are inserted prior to the processing of each positive batch and are limited to the allowed perturbation budget. The *historical baseline* considers historical negative edges as defined in (Poursafaei et al., 2022), meaning edges that were observed before but were not present at the current time. These negative edges are challenging for TGNN models as they were encountered previously and required temporal reasoning from the model to clearly distinguish them from the real edges. Therefore by inserting them as negatives, the model might be more prune to the attack.

**Memstranding settings.** MemStranding Dai et al. (2023) represents a sophisticated evasion attack specifically designed for temporal graph neural networks that leverages memory-based architectures. Unlike our TR-BCD approach that distributes adversarial edges across temporal batches, MemStranding operates as a single-shot evasion attack that inserts a burst of fake edges at a single timestamp to corrupt node memory states. The attack identifies high-degree victim nodes and their neighbors, then strategically injects fake messages at a selected timestamp to manipulate their memory states. MemStranding simulates fake neighbors by sampling from Gaussian distributions based on the standard deviation of current neighbor memory vectors, creating target noisy memory states that degrade model performance. The attack is persistent, affecting all future predictions after the injection timestamp. In our experimental evaluation, we integrate MemStranding as a strong baseline with a 5% attack budget, demonstrating that while it can achieve significant performance degradation in some cases (notably achieving the best performance on TGN-MOOC with 0.0949 MRR), our TR-BCD method consistently outperforms it across most model-dataset combinations, highlighting the advantages of our distributed gradient-based approach over single-shot burst attacks. Due to the lack of publicly available code and since the authors did not provide a copy upon request, we re-implemented their attack for our experiments.

**Memory Requirements for TR-BCD.** If we assume 4 bytes per parameter and the default batch size of 200, then storing the parameters alone for Reddit requires $4\,\text{B} \cdot 10,984^2 \cdot 15\% \cdot 672,447/200 \approx 243\,\text{GB}$. Here, we assume that for each benign test batch, we may choose the perturbations from the $10,984 \times 10,984$ edges of Reddit (no duplicates allowed within a single adversarial batch).

## D    CONTEXTUAL PERTURBATION ON TEST EDGES

Table 5: Feature perturbation attack (using FGSM) results for temporal link prediction on attributed CTDG datasets with $\epsilon = 0.3$. Performance is reported in Mean Reciprocal Rank (MRR), averaged over five trials.

| Model | Attack | Wikipedia | Reddit | MOOC |
|---|---|---|---|---|
| TGN | None | $0.3929_{\pm0.0366}$ | $0.4550_{\pm0.0485}$ | $0.1574_{\pm0.0590}$ |
| | FGSM | $\mathbf{0.2638}_{\pm0.0388}$ | $\mathbf{0.4518}_{\pm0.0444}$ | $\mathbf{0.1569}_{\pm0.0581}$ |
| TNCN | None | $0.7207_{\pm0.0009}$ | $0.7228_{\pm0.0064}$ | $0.2521_{\pm0.0192}$ |
| | FGSM | $\mathbf{0.6944}_{\pm0.0230}$ | $\mathbf{0.6559}_{\pm0.0492}$ | $\mathbf{0.2475}_{\pm0.0235}$ |

Our method attacks the victim models primarily through structural perturbations as outlined in Section 4. In addition, we explore the effect of applying contextual perturbations to the features of the adversarial edges. As detailed in Section 5, incorporating contextual perturbations on these edges results in little to no additional degradation in test performance, suggesting that the models are mainly vulnerable to structural changes. To further validate our findings, we also experiment with applying contextual attacks directly on the test edges. In this setup, we inject a small amount of noise, crafted via FGSM (Goodfellow, 2014), into the feature space of the test data. Our experiments reveal that applying FGSM-based contextual perturbations to the edge features produces varied effects across models and datasets. For instance, while TGN on Wikipedia experiences a noticeable drop in MRR when subjected to FGSM, the impact on TNCN and on other datasets such as Reddit and MOOC

remains minimal. These results suggest that, although direct feature perturbations can influence performance in certain cases, the dominant vulnerability stems from structural perturbations.

# E  ABLATION STUDY ON PERTURBATION CONTRIBUTIONS

While our main experiments demonstrate the effectiveness of TR-BCD across different datasets and models, understanding the individual contribution of each perturbation type is crucial for developing targeted defense strategies. To this end, we conduct an extended ablation study focusing on the contribution of individual perturbations to the overall attack performance.

**Experimental Setup.** We evaluate the impact of different perturbation combinations on both TGN and TNCN models across three representative datasets: Wikipedia, Subreddit, and MOOC. These datasets were selected to provide diversity in terms of graph structure (bipartite vs. non-bipartite), temporal dynamics, and feature availability. The perturbation variations are defined as follows:

- **Structural Only**: TR-BCD for edge selection, timestamps chosen randomly within the dataset's time range, random valid features sampled from the dataset.

- **Structural + Temporal**: Current TR-BCD setting with TR-BCD for node pair selection, Gaussian sampling for timestamps, valid features from dataset.

- **Structural + Contextual**: TR-BCD for node pair selection, random timestamp selection, FGSM applied to valid edge features.

- **All Perturbations**: Complete TR-BCD implementation.

**Results and Analysis.** Table 6 presents the comprehensive ablation results across all perturbation combinations. The results reveal several important insights about the relative effectiveness of different perturbation types.

Table 6: Extended ablation study results showing the contribution of individual perturbation types to attack performance. Performance is measured in Mean Reciprocal Rank (MRR) averaged over 5 trials. Bold values indicate the best performing attack for each model-dataset combination.

| Model | Perturbations | Wikipedia | Subreddit | MOOC |
|---|---|---|---|---|
| TGN | No Attack | $0.3929_{\pm 0.0366}$ | $0.4550_{\pm 0.0485}$ | $0.1574_{\pm 0.0590}$ |
| | Structural Only | $0.2504_{\pm 0.0618}$ | $\mathbf{0.3037}_{\pm 0.1091}$ | $0.1320_{\pm 0.0554}$ |
| | Structural+Temporal | $0.2233_{\pm 0.0546}$ | $0.3350_{\pm 0.1191}$ | $\mathbf{0.1245}_{\pm 0.0491}$ |
| | Structural+Contextual | $0.2538_{\pm 0.0552}$ | $0.3233_{\pm 0.1124}$ | $0.1266_{\pm 0.0480}$ |
| | All Perturbations | $\mathbf{0.2073}_{\pm 0.0597}$ | $0.3403_{\pm 0.1063}$ | $0.1273_{\pm 0.0423}$ |
| TNCN | No Attack | $0.7207_{\pm 0.0009}$ | $0.7228_{\pm 0.0064}$ | $0.2521_{\pm 0.0192}$ |
| | Structural Only | $0.7050_{\pm 0.0099}$ | $0.6404_{\pm 0.1009}$ | $\mathbf{0.1118}_{\pm 0.0285}$ |
| | Structural+Temporal | $0.7057_{\pm 0.0167}$ | $0.5869_{\pm 0.1688}$ | $0.1164_{\pm 0.0236}$ |
| | Structural+Contextual | $0.7082_{\pm 0.0113}$ | $0.6123_{\pm 0.1368}$ | $0.1138_{\pm 0.0275}$ |
| | All Perturbations | $\mathbf{0.7043}_{\pm 0.0097}$ | $\mathbf{0.5807}_{\pm 0.1802}$ | $0.1149_{\pm 0.0297}$ |

The ablation results reveal several important patterns that provide deeper insights into the vulnerability landscape of temporal graph neural networks:

**Dominance of Structural Perturbations.** Structural perturbations alone demonstrate remarkable effectiveness, sometimes achieving superior performance compared to combinations with other perturbation types. This finding is particularly evident for TGN on the Subreddit dataset, where structural-only attacks achieve the best performance ($0.3037 \pm 0.1091$), outperforming even the complete attack combination. This suggests that the topological structure of temporal graphs represents the primary attack surface for adversarial perturbations.

**Complementary Effects of Perturbation Types.** While structural perturbations form the foundation of effective attacks, the combination of all perturbation types yields the strongest attack in approximately half of the cases. For TGN on Wikipedia and TNCN on both Wikipedia and Subreddit, the complete attack achieves optimal performance. This indicates that while structural perturbations are

necessary, temporal and contextual perturbations can provide complementary benefits that enhance overall attack effectiveness.

**Dataset-Model Interplay** The results reveal a complex interplay between dataset characteristics and model vulnerabilities. Each dataset-model combination exhibits distinct susceptibility patterns:

- **TGN on Wikipedia**: Benefits most from the complete attack combination, suggesting this model-dataset pair is vulnerable to coordinated multi-dimensional perturbations
- **TGN on Subreddit**: Most vulnerable to structural-only attacks, indicating that temporal and contextual perturbations may introduce noise that reduces attack effectiveness
- **TNCN on MOOC**: Shows optimal vulnerability to structural-only attacks, highlighting the importance of graph topology for this particular combination

**Temporal vs. Contextual Perturbations** The comparison between structural+temporal and structural+contextual perturbations reveals interesting patterns. Temporal perturbations appear to be more effective for TGN on Wikipedia and MOOC datasets, while contextual perturbations show mixed results. This suggests that temporal dynamics play a more critical role in determining model vulnerability than feature perturbations, particularly for models that rely heavily on temporal reasoning.

**Implications for Defense Strategies** These findings have important implications for developing robust temporal graph neural networks:

- **Prioritize Structural Defense**: Given the dominance of structural perturbations, defense mechanisms should prioritize protecting graph topology integrity
- **Model-Specific Vulnerabilities**: Different models exhibit varying susceptibility patterns, suggesting the need for model-specific defense strategies
- **Dataset-Dependent Robustness**: The varying effectiveness across datasets indicates that robustness evaluation should consider multiple graph types and domains
- **Multi-Dimensional Defense**: While structural defense is primary, comprehensive defense strategies should address temporal and contextual perturbations as well

This extended ablation study provides crucial insights into the relative contributions of different perturbation types in adversarial attacks on temporal graph neural networks. The results demonstrate that while structural perturbations form the foundation of effective attacks, the optimal attack strategy varies significantly across different model-dataset combinations. These findings underscore the importance of developing comprehensive defense strategies that address multiple attack vectors while recognizing the dataset-model specific nature of vulnerabilities in temporal graph learning systems.

# F  SENSITIVITY ANALYSIS

## F.1  THE EFFECT OF BLOCK SIZE

We study the impact of varying the block size parameter $b$ in our TR-BCD attack algorithm on attack effectiveness across different datasets and temporal graph neural network (TGNN) models. The block size in TR-BCD denotes the size of the sample space of edge candidates that are randomly selected and optimized at each iteration of the attack algorithm. By limiting gradient-based updates and memory usage to a block of size $b$, this parameter enables the attack to scale efficiently in large temporal graphs, while maintaining the ability to select adversarial edges from a sufficiently diverse pool. Larger block sizes allow for higher attack effectiveness due to greater candidate diversity, but incur higher computational cost per iteration, whereas small blocks increase efficiency but may reduce attack strength by limiting the solution space explored. We evaluate four block sizes ranging from 100 to 100000, measuring the mean reciprocal rank (MRR) drop averaged over 3 evaluation runs. The results in Table 7 are reported for two model groups: Temporal Graph Network (TGN) and Temporal Neural Common Neighbor (TNCN), across the three datasets: Wikipedia, Reddit, and MOOC. The results show that the strength of the attack generally increases with block size before

Table 7: MRR drop (↓) under TR-BCD attack for varying block sizes. Results are reported for TGN and TNCN models across Wiki, Reddit, and MOOC datasets.

| Model | Block Size | Wikipedia | Reddit | MOOC |
|---|---|---|---|---|
| TGN | 100 | $0.3748 \pm 0.0262$ | $0.4375 \pm 0.0442$ | $0.1359 \pm 0.0367$ |
| | 1000 | $0.3072 \pm 0.0290$ | $0.4261 \pm 0.0499$ | $0.1288 \pm 0.0287$ |
| | 10000 | $\mathbf{0.2533} \pm 0.0674$ | $0.3433 \pm 0.1499$ | $0.1169 \pm 0.0237$ |
| | 100000 | $0.2892 \pm 0.0735$ | $\mathbf{0.2474} \pm 0.1631$ | $\mathbf{0.0908} \pm 0.0053$ |
| TNCN | 100 | $0.7200 \pm 0.0003$ | $0.7219 \pm 0.0070$ | $0.2193 \pm 0.0082$ |
| | 1000 | $0.7187 \pm 0.0007$ | $0.7246 \pm 0.0018$ | $0.1898 \pm 0.0100$ |
| | 10000 | $0.7122 \pm 0.0028$ | $0.6661 \pm 0.0764$ | $\mathbf{0.1149} \pm 0.0118$ |
| | 100000 | $\mathbf{0.6998} \pm 0.0029$ | $\mathbf{0.4929} \pm 0.1246$ | $0.1161 \pm 0.0146$ |

plateauing. In most cases, the largest block size corresponds to the strongest attack. Exceptions are limited to two cases, where the attack MRR at the largest block size remains within one standard deviation of the best-observed attack outcome.

## F.2 THE EFFECT OF $\epsilon$ IN CONTEXTUAL PERTURBATION

In this section, we investigate how the intensity of contextual perturbation, parameterized by $\epsilon$ in the FGSM attack, affects the robustness of Temporal Graph Neural Networks (TGNNs) under adversarial edge feature modification. The FGSM-based contextual attack perturbs edge features of positive edges, with $\epsilon$ controlling the maximum $L_\infty$ norm of the perturbation for each feature dimension. Larger values of $\epsilon$ allow for greater changes in features, typically resulting in more effective attacks (Goodfellow, 2014). Table 8 shows that increasing the $\epsilon$ generally strengthens the contextual attack, resulting in lower MRR scores on Wikipedia and Reddit datasets for both TGN and TNCN models. The highest value $\epsilon = 1.0$ consistently produces the lowest MRR, except for the MOOC data set, where there is little change, possibly due to the limited variance in its edge features. This emphasizes that attack strength via feature perturbation is highly dependent on both the chosen $\epsilon$ and underlying dataset properties.

Table 8: MRR drop (↓) for varying $\epsilon$ in FGSM contextual perturbation on positive edges, across TGN and TNCN models and three datasets. The lowest mean MRR in each column is marked in **bold**.

| Model | $\epsilon$ | Wiki | Reddit | Mooc |
|---|---|---|---|---|
| TGN | 0 | $0.4105 \pm 0.0195$ | $0.4719 \pm 0.0471$ | $0.1428 \pm 0.0503$ |
| | 0.1 | $0.3872 \pm 0.0212$ | $0.4756 \pm 0.0445$ | $0.1424 \pm 0.0500$ |
| | 0.3 | $0.2999 \pm 0.0269$ | $0.4798 \pm 0.0367$ | $\mathbf{0.1417} \pm 0.0486$ |
| | 0.7 | $0.1657 \pm 0.0335$ | $0.4553 \pm 0.0376$ | $0.1423 \pm 0.0461$ |
| | 1.0 | $\mathbf{0.1103} \pm 0.0232$ | $\mathbf{0.4174} \pm 0.0456$ | $0.1423 \pm 0.0432$ |
| TNCN | 0 | $0.7212 \pm 0.0009$ | $0.7264 \pm 0.0014$ | $0.2439 \pm 0.0154$ |
| | 0.1 | $0.7187 \pm 0.0013$ | $0.7238 \pm 0.0022$ | $0.2425 \pm 0.0174$ |
| | 0.3 | $0.7007 \pm 0.0175$ | $0.6929 \pm 0.0113$ | $0.2396 \pm 0.0200$ |
| | 0.7 | $0.4690 \pm 0.1670$ | $0.1132 \pm 0.0219$ | $0.2334 \pm 0.0241$ |
| | 1.0 | $\mathbf{0.3486} \pm 0.1709$ | $\mathbf{0.0424} \pm 0.0059$ | $\mathbf{0.2266} \pm 0.0264$ |

## G EDGE INSERTION VS. DELETION

While our main approach focuses on adversarial edge insertion attacks, an alternative strategy involves edge deletion, where existing edges are removed from the temporal graph. To explore this complementary attack surface, we adapted TR-BCD to perform edge deletion by treating the real edges in each batch as candidates and "flipping" selected edge weights from 1 to 0, effectively removing them before the memory-update step. This approach targets the models by limiting the

information flow within the temporal graph, potentially degrading the established temporal patterns and node relationships in the model memory.

Table 9 presents results comparing edge deletion attacks with our best insertion-based method across three datasets and the two TGNN architectures. The results reveal interesting patterns in model vulnerability to different perturbation strategies.

Table 9: Comparison of edge deletion and insertion attacks on temporal link prediction. Performance is reported in Mean Reciprocal Rank (MRR) averaged over 5 trials. Bold values indicate the best attack performance for each model-dataset combination.

| Model | Attack Strategy | Wikipedia | Enron | UCI |
|---|---|---|---|---|
| TGN | No Attack | $0.3929 _{\pm 0.0366}$ | $0.4257 _{\pm 0.0123}$ | $0.3058 _{\pm 0.0104}$ |
| | Edge Deletion (TR-BCD) | $0.3708 _{\pm 0.0428}$ | $\mathbf{0.2294} _{\pm 0.0190}$ | $\mathbf{0.2000} _{\pm 0.0525}$ |
| | Best Insertion | $\mathbf{0.1934} _{\pm 0.0587}$ | $0.2321 _{\pm 0.0229}$ | $0.2857 _{\pm 0.0181}$ |
| TNCN | No Attack | $0.7207 _{\pm 0.0009}$ | $0.4257 _{\pm 0.0123}$ | $0.4839 _{\pm 0.0030}$ |
| | Edge Deletion (TR-BCD) | $0.7060 _{\pm 0.0036}$ | $0.4248 _{\pm 0.0144}$ | $\mathbf{0.4456} _{\pm 0.0023}$ |
| | Best Insertion | $\mathbf{0.7021} _{\pm 0.0137}$ | $\mathbf{0.3980} _{\pm 0.0232}$ | $0.4793 _{\pm 0.0025}$ |

The comparative analysis reveals that both attack strategies demonstrate effectiveness, but with notable variations across datasets and architectures.

The comparative analysis reveals that both attack strategies demonstrate effectiveness, but with notable variations across datasets and architectures. For TGN models, edge deletion attacks prove particularly effective on the Enron and UCI datasets, achieving substantial performance degradation. However, on Wikipedia, the insertion-based approach maintains superiority, reducing MRR by over 50%. TNCN models show greater resilience to edge deletion attacks overall, with insertion-based methods consistently achieving better or comparable attack performance across all datasets. While not the main focus of this work, future work should consider applying our TR-BCD in a setting of simultaneous edge insertion and deletion, as each exploits different vulnerabilities in temporal graph neural networks and due to TR-BCD's straightforward adaptation to this joint setting.

