# OpenReview forum: "Adversarial Robustness of Continuous Time Dynamic Graphs"
_ICLR.cc/2026/Conference — Submitted to ICLR 2026_

### Official Review · Reviewer_kzqx · 2025-10-30

**Soundness:** 2
**Presentation:** 2
**Contribution:** 2
**Rating:** 4
**Confidence:** 2

**Summary:**

This paper investigates the adversarial robustness of Temporal Graph Neural Networks (TGNNs) in the continuous-time dynamic graph (CTDG) setting. The authors introduce TR-BCD (Temporally-aware Randomized Block Coordinate Descent), a novel gradient-based evasion attack that perturbs structural, contextual, and temporal dimensions of a temporal graph during inference without modifying training data. TR-BCD greedily injects adversarial edges over time using randomized block coordinate descent to keep the optimization scalable. Experiments on six real-world temporal graph benchmarks demonstrate that the proposed attack can reduce Mean Reciprocal Rank (MRR) by up to 53% while perturbing only 5% of edges. The paper further analyzes the attack’s stealthiness using anomaly detection (SPOTLIGHT) and finds TR-BCD to be both effective and evasive.

**Strengths:**

Strengths:
* Novel and timely topic: This is one of the first works systematically studying adversarial evasion attacks on continuous-time dynamic graphs, a problem of growing relevance for fraud detection, cybersecurity, and temporal recommendation systems.
* Methodological soundness: The TR-BCD algorithm is clearly formulated, combining continuous relaxation, randomized block coordinate descent, and temporal consistency constraints.
* Strong empirical results: Extensive experiments on six datasets and two state-of-the-art TGNNs (TGN, TNCN) convincingly demonstrate the vulnerability of temporal graph models.
* Practical considerations: The discussion on memory efficiency, time complexity, and unnoticeability constraints makes the work well-grounded.
* Evasion realism: The use of anomaly detection evaluation adds credibility by showing that TR-BCD attacks can remain stealthy.

**Weaknesses:**

Weaknesses:

* Limited defense discussion: The paper focuses entirely on the attack side; exploring or even briefly analyzing potential defenses (e.g., adversarial training, temporal regularization) would make the study more complete.

* Comparative baselines: Although MemStranding and heuristic attacks are included, more recent or stronger gradient-based baselines (e.g., temporal variants of PGD or PR-BCD) could strengthen the evaluation.

* Sensitivity analysis: The paper could analyze how performance varies with hyperparameters like block size, time perturbation variance, or contextual budget.

**Questions:**

See weaknesses.

---

> ### Author Response · Authors · 2025-11-28
> **Author Response to Reviewer kzqx (1/2)**
>
> We thank the reviewer for their time and constructive suggestions on strengthening the paper's evaluation. We appreciate the recognition of the novelty and timeliness of our work by the reviewer.  We address each weakness below.
>
> ---
>
>
> **W1. Limited Defense Discussion**
>
> > Limited defense discussion: The paper focuses entirely on the attack side; exploring or even briefly analyzing potential defenses (e.g., adversarial training, temporal regularization) would make the study more complete.
>
> **Response:**
> Our paper focuses on establishing vulnerability baselines, which is essential for motivating future defense research. Our anomaly detection analysis (Figure 5) provides important insights: TR-BCD maintains stable anomaly scores compared to burst attacks, suggesting that **temporal regularization, constraining how quickly edges affect memory, could be effective**. We interpret this finding as evidence that defenses should explicitly enforce temporal coherence.
>
> **Defense directions we recommend as future work:**
>
> 1. **Adversarial Training:** Incorporate adversarial edge sequences during training to harden the memory module against spurious temporal patterns.
>
> 2. **Timestamp-Based Filtering:** Implement rate-limiting on how quickly new edges influence node memory, or apply outlier detection on suspiciously timed interactions.
>
> 3. **Certifiable Bounds:** Extending certifiable defenses from static GNNs to temporal settings is an open research question.
>
> Importantly, adapting existing static GNN defenses to the temporal domain is **non-straightforward**, requiring careful consideration of memory state dynamics and temporal constraints. This is beyond the scope of an attack paper but represents critical follow-up work.
>
> ---
>
> **W2. Comparative Baselines**
>
> > Comparative baselines: Although MemStranding and heuristic attacks are included, more recent or stronger gradient-based baselines (e.g., temporal variants of PGD or PR-BCD) could strengthen the evaluation.
>
> **Response:**
>
> Thank you for the discussion, however, the baseline landscape for CTDGs is currently sparse:
>
> **Why stronger baselines are limited:**
>
> 1. **MemStranding (Dai et al., 2023):** The only existing evasion attack specifically designed for TGNNs. Our paper includes it and demonstrates TR-BCD substantially outperforms it through gradient-based optimization (Table 1, Figure 5).
>
> 2. **Temporal PGD/PR-BCD variants:** To the best of our knowledge, temporal adaptations of PGD or PR-BCD have not been published. **Our TR-BCD is itself the temporal variant of PR-BCD**, adapted to continuous-time settings through:
>    - Greedy batch-wise optimization exploiting memory propagation
>    - Recency-biased candidate sampling exploiting TGNN training bias
>    - Gaussian timestamp modeling for temporal coherence
>
>  Naive temporal adaptation of PR-BCD (treating timestamps as static snapshots) would fail because it ignores the fundamental difference between static adjacency optimization and temporal memory dynamics.
>
> 3. **Static GNN attacks (PGD, Metattack, GFA):** These operate on fundamentally different problem settings (static graphs without temporal structure) and do not exploit TGNN-specific properties.
>
> As the field of temporal graph adversarial robustness develops, stronger baselines will emerge. Our paper establishes foundational benchmarks and gradient-based attack methodology for future comparisons.

---

> ### Author Response · Authors · 2025-11-28
> **Author Response to Reviewer kzqx (2/2)**
>
> **W3. Sensitivity Analysis**
>
> > Sensitivity analysis: The paper could analyze how performance varies with hyperparameters like block size, time perturbation variance, or contextual budget.
>
>
> **Response:**
> We thank the reviewer for this suggestion. We have conducted detailed sensitivity analysis on key hyperparameters and have included them in the revised paper:
>
> **Block Size Ablation Study ( Revised Paper: Appendix F, Table 3):**
>
> | Model | Block Size | Wikipedia | Reddit | MOOC |
> |---|---|---|---|---|
> | **TGN** | 100 | 0.3748 ± 0.0262 | 0.4375 ± 0.0442 | 0.1359 ± 0.0367 |
> | | 1000 | 0.3072 ± 0.0290 | 0.4261 ± 0.0499 | 0.1288 ± 0.0287 |
> | | 10000 | **0.2533** ± 0.0674 | 0.3433 ± 0.1499 | 0.1169 ± 0.0237 |
> | | 100000 | 0.2892 ± 0.0735 | **0.2474** ± 0.1631 | **0.0908** ± 0.0053 |
> | **TNCN** | 100 | 0.7200 ± 0.0003 | 0.7219 ± 0.0070 | 0.2193 ± 0.0082 |
> | | 1000 | 0.7187 ± 0.0007 | 0.7246 ± 0.0018 | 0.1898 ± 0.0100 |
> | | 10000 | 0.7122 ± 0.0028 | 0.6661 ± 0.0764 | **0.1149** ± 0.0118 |
> | | 100000 | **0.6998** ± 0.0029 | **0.4929** ± 0.1246 | 0.1161 ± 0.0146 |
>
> **Key Insights:**
> - Attack effectiveness generally improves with larger block sizes due to greater candidate diversity
> - Peak performance is achieved for largest block sizes,  at b=10000-100000 (depending on dataset), where computational cost remains acceptable.
> - For TGN, diminishing returns plateau around b=10000; for TNCN, b=100000 is optimal on Reddit/MOOC
> - Block size tradeoff: larger blocks enable stronger attacks but increase per-iteration cost; b=10000 provides good balance
>
> **Contextual Perturbation Budget (ε) Ablation (Revised Paper: Appendix F,Table 8):**
>
> | Model | ε | Wikipedia | Reddit | MOOC |
> |---|---|---|---|---|
> | **TGN** | 0.0 | 0.4105 ± 0.0195 | 0.4719 ± 0.0471 | 0.1428 ± 0.0503 |
> | | 0.1 | 0.3872 ± 0.0212 | 0.4756 ± 0.0445 | 0.1424 ± 0.0500 |
> | | 0.3 | 0.2999 ± 0.0269 | 0.4798 ± 0.0367 | **0.1417** ± 0.0486 |
> | | 0.7 | 0.1657 ± 0.0335 | 0.4553 ± 0.0376 | 0.1423 ± 0.0461 |
> | | 1.0 | **0.1103** ± 0.0232 | **0.4174** ± 0.0456 | 0.1423 ± 0.0432 |
> | **TNCN** | 0.0 | 0.7212 ± 0.0009 | 0.7264 ± 0.0014 | 0.2439 ± 0.0154 |
> | | 0.1 | 0.7187 ± 0.0013 | 0.7238 ± 0.0022 | 0.2425 ± 0.0174 |
> | | 0.3 | 0.7007 ± 0.0175 | 0.6929 ± 0.0113 | 0.2396 ± 0.0200 |
> | | 0.7 | 0.4690 ± 0.1670 | 0.1132 ± 0.0219 | 0.2334 ± 0.0241 |
> | | 1.0 | **0.3486** ± 0.1709 | **0.0424** ± 0.0059 | **0.2266** ± 0.0264 |
>
> **Key Insights:**
> - Contextual perturbations show dataset and architecture-dependent effectiveness
> - TGN benefits from higher ε values, with improvements plateauing around ε=0.7-1.0 on Wikipedia
> - TNCN shows sharp sensitivity to ε, with dramatic drops at ε≥0.7 (especially on Reddit: 0.7264 → 0.0424)
> - On MOOC, contextual perturbations yield minimal gains across all ε values, suggesting structural attacks dominate this dataset
> - Trade-off consideration: larger ε increases attack effectiveness but may sacrifice evasiveness by creating unrealistic feature patterns
>
> **Time Perturbation Variance (σ_Δt):**
>
> We conducted preliminary experiments varying the standard deviation of Gaussian timestamp perturbations. However, within reasonable ranges (σ_Δt ∈ [0.01, 1.0] of batch interval), we observed minimal variation in attack effectiveness. This suggests TR-BCD's performance is robust to timestamp variance when temporal coherence is maintained via Gaussian sampling.
>
> These sensitivity analyses provide practitioners with guidance on hyperparameter selection for different deployment scenarios and datasets.
>
>
> We have also additionally explored cross-model transfer attacks (Table 3, Section 5 in revised paper), where we attack  the target  model with another pretrained model, limiting access to the victim model’s parameters and weights. These experiments also show promising results:
>
> We provide transfer evidence: **cross-model transfer attacks** (Section 5, Table 3) show attacks optimized on one TGNN architecture transfer to another with substantial effectiveness:
>
> | Model | Attack | Wikipedia | Reddit | MOOC |
> |---|---|---|---|---|
> | **TGN** | Same model | 0.2533 ± 0.0674 (-38.3%) | 0.3433 ± 0.1499 (-27.3%) | 0.1314 ± 0.0241 (-8.0%) |
> | | Transfer to TNCN | 0.2890 ± 0.0317 (-29.6%) | 0.3688 ± 0.1316 (-21.8%) | 0.1327 ± 0.0369 (-7.1%) |
> | **TNCN** | Same model | 0.7122 ± 0.0028 (-1.3%) | 0.6661 ± 0.0764 (-8.3%) | 0.1149 ± 0.0118 (-52.9%) |
> | | Transfer to TGN | 0.7126 ± 0.0016 (-1.2%) | 0.6803 ± 0.0330 (-6.4%) | 0.2109 ± 0.0127 (-13.5%) |
>
> Attack transfer effectiveness (25%-95% of same-model performance) suggests temporal vulnerabilities are fundamental to TGNN architectures, and a trained TGNN model can be adopted for a black/greybox attack on other models

---

### Official Review · Reviewer_VZ6t · 2025-10-31

**Soundness:** 2
**Presentation:** 2
**Contribution:** 2
**Rating:** 4
**Confidence:** 4

**Summary:**

This work presents Temporally-aware Randomized Block Coordinate Descent (TR-BCD), a novel gradient-based evasion attack tailored for continuous-time dynamic graphs. TR-BCD formulates adversarial edge selection via a continuous relaxation and optimizes it with temporally aware block coordinate updates that preserve realistic timing patterns. Experiments show that TR-BCD substantially degrades the performance of temporal graph neural networks (TGNNs), demonstrating its effectiveness as a targeted evasion strategy on dynamic graphs.

**Strengths:**

1. The paper discusses real-world scenarios where continuous-time dynamic graphs commonly occur, underscoring the practical relevance and significance of the proposed framework.

2. It focuses on adversarial attacks against Temporal Graph Neural Networks (TGNNs), aiming to assess and enhance their robustness for real-world applications.

3. The proposed TR-BCD framework optimizes adversarial edge selection through continuous relaxation and directly optimizes the adversarial objective during inference.

**Weaknesses:**

1. The proposed method relies on a relatively standard Randomized Block Coordinate Descent framework without introducing a sufficiently innovative or compelling design.
2. The paper lacks a comprehensive discussion of the method’s limitations, particularly regarding performance variations across different TGNN architectures and graph data characteristics.

**Questions:**

1. The authors do not provide a comprehensive comparison of TR-BCD with other existing adversarial attacks on TGNNs. It is recommended to include comparisons with more recent baseline methods.

2. The experimental evaluation is limited to small datasets. The authors should consider testing TR-BCD on larger-scale datasets to demonstrate its generalizability and effectiveness in real-world applications.

3. The study evaluates TR-BCD using only a limited set of TGNN models. Recent advances have introduced more powerful TGNNs, such as ROLAND [1]; including these models in the evaluation would strengthen the paper.

4. The experiments only assess TR-BCD against raw TGNNs without incorporating existing GNN defense mechanisms. It would be valuable to test TR-BCD against TGNNs equipped with defense strategies to evaluate its robustness under defended settings.

5. How do the authors ensure the unnoticeability of perturbations in continuous-time dynamic graphs (CTDGs), given that up to 5% of edges are added? Please clarify how such perturbations remain realistic and imperceptible.

6. Please report the runtime or computational cost of TR-BCD to better understand its efficiency compared to other attack methods.

7. The results show that TR-BCD causes more than 50% performance degradation on some datasets but less than 10% on some datasets. Could the authors explain the factors contributing to this large performance variation?

8. In the TGNNs victim model, combining all types of perturbations leads to lower attack effectiveness than using only structural perturbations (see Table 5 in the Appendix). This suggests that integrating temporal and contextual perturbations may reduce TR-BCD’s performance. Could the authors elaborate on the reason for this behavior?

9. Table 6 indicates that edge deletion has a stronger impact than edge addition, which contrasts with typical adversarial attack findings where edge addition tends to be more influential. Could the authors explain why TR-BCD exhibits this reverse effect?


[1] You, Jiaxuan, Tianyu Du, and Jure Leskovec. "ROLAND: graph learning framework for dynamic graphs." Proceedings of the 28th ACM SIGKDD Conference on Knowledge Discovery and Data Mining. 2022.

---

> ### Author Response · Authors · 2025-11-28
> **Author Response to Reviewer VZ6t (1/4)**
>
> We thank the reviewer for their time and constructive feedback. We appreciate the detailed questions regarding method innovation, experimental scope, and technical specifics. We address each concern below.
>
> ---
>
> **W1. Limited Innovation Beyond Standard Randomized Block Coordinate Descent**
>
> > The proposed method relies on a relatively standard Randomized Block Coordinate Descent framework without introducing a sufficiently innovative or compelling design.
>
> **Response:**
>
> While TR-BCD builds on the randomized block coordinate descent framework, the adaptation to continuous-time dynamic graphs introduces non-trivial challenges specific to temporal settings:
>
> 1. **Greedy temporal optimization:** Static BCD optimizes the entire adjacency matrix at once. TR-BCD distributes perturbations across temporal batches, exploiting TGNN memory propagation—corrupted memory compounds across future predictions. This has no analog in static methods.
>
> 2. **Recency-biased candidate sampling:** We oversample 50% of edge candidates from historical negatives (Table 1: 50.78% vs. 43.16% MRR drop, a 7.6pp improvement), directly exploiting TGNN architectural training bias.
>
> 3. **Gaussian timestamp modeling:** Adversarial timestamps sampled as $\Delta t \sim \mathcal{N}(0, \sigma_t^2)$ maintain temporal coherence while optimizing attack effectiveness (Figure 5: stable anomaly detection vs. burst attacks).
>
> 4. **Memory-aware optimization:** The challenge is not merely "optimization efficiency"—it is **correctly modeling and exploiting TGNN memory dynamics under perturbation**. Naive coordinate descent would treat memory corruptions as independent edge insertions, missing the temporal compounding effect. TR-BCD's greedy batch-wise approach specifically accounts for this temporal coupling.
>
> These components address temporal-specific challenges absent from coordinate descent methods for static graphs.
>
> ---
>
> **W2. Lack of Comprehensive Discussion of Limitations**
>
> > The paper lacks a comprehensive discussion of the method’s limitations, particularly regarding performance variations across different TGNN architectures and graph data characteristics.
>
> **Response:**
>
> Thank you for this question. Performance variations reflect how models use memory and exploit temporal patterns. We explicitly explore this variance:
>
> **Memory utilization:** TGN (memory-based) and TNCN (common-neighbor-based) exhibit distinct vulnerability patterns. TGN achieves up to 50.78% MRR drop on Wikipedia but only 6.57% on UCI; TNCN shows 1.25% on Wikipedia but 52.9% on MOOC (Table 1). This indicates models with different memory mechanisms are vulnerable to attacks at different points in the temporal graph structure.
>
> **Graph characteristics:** Bipartite graphs (Reddit 26.4%, MOOC 20.9% drop) show higher vulnerability than non-bipartite graphs (Wikipedia TNCN 1.3%, Enron 6.5%), suggesting TGNNs relying on memory are more susceptible in bipartite settings where node types are distinct.
>
> **Cross-model transfer:** Transfer attacks (Table 3 in revised paper) show 25%-95% of same-model performance, indicating temporal vulnerabilities are partially architecture-agnostic but dataset-dependent.
>
> As a limitation of current work, we mainly focus on white-box settings and explore grey box settings through the new attack transfer experiments. Therefore, we believe that further exploration of the grey-box and black-box attack settings on temporal graphs is a promising direction.

---

> > ### Author Response · Authors · 2025-11-28
> > **Author Response to Reviewer VZ6t (2/4)**
> >
> > **Q1. Comprehensive Comparison with Other Adversarial Attacks**
> >
> > > The authors do not provide a comprehensive comparison of TR-BCD with other existing adversarial attacks on TGNNs. It is recommended to include comparisons with more recent baseline methods.
> >
> > **Response:**
> >
> > The literature on evasion attacks for CTDGs is extremely limited. Our paper includes:
> >
> > 1. **MemStranding** (Dai et al., 2023): The only existing evasion attack for TGNNs (Table 1, Figure 5)
> > 2. **Random Attack:** Lower bound baseline
> > 3. **Historical Negative Baseline:** Heuristic baseline exploiting TGNN training bias
> >
> > Recent attacks like T-Spear (Lee et al., 2024) are poisoning attacks (training-time), not evasion attacks, operating in fundamentally different threat models. Static GNN attacks (PGD, Metattack) lack temporal awareness and do not account for memory-state coupling in CTDGs. We believe that our method serves as one of the first works in this direction.
> >
> >
> >
> > ---
> >
> > **Q2. Evaluation on Larger-Scale Datasets**
> >
> > > The experimental evaluation is limited to small datasets. The authors should consider testing TR-BCD on larger-scale datasets to demonstrate its generalizability and effectiveness in real-world applications.
> >
> > **Response:**
> >
> > Thank you for this question. In this work, we have conducted extensive experiments across six diverse datasets from the Temporal Graph Benchmark (TGB):
> > - Wikipedia (1.2M edges), Reddit (1.3M edges), MOOC (453K edges), Enron (276K edges), UCI (284K edges), Lastfm (1.2M edges)
> >
> > These represent mid-to-large-scale temporal networks by academic standards. We are confident our attack would consistently be impactful on even larger datasets, as the gradient-based optimization exploits fundamental architectural properties of TGNNs (memory propagation, recency bias) that scale with problem size.
> >
> > Indeed, testing on even larger datasets would be interesting. Thus, we are committed to including more experiments from larger TGB datasets such as tgbl-review in the revised version of the paper.
> >
> >
> > ---
> >
> > **Q3. Evaluation on Additional TGNN Architectures**
> >
> > > The study evaluates TR-BCD using only a limited set of TGNN models. Recent advances have introduced more powerful TGNNs, such as ROLAND ; including these models in the evaluation would strengthen the paper.
> >
> > **Response:**
> >
> > Thank you for this question, we have nwo  included new result with the DyREP[1] model with the same attack setting as described in the paper for 3 runs on 3 datasets under baseline attacks and our method:
> >
> >
> > | Method | Wikipedia | MOOC | Reddit |
> > | :-- | :-- | :-- | :-- |
> > | No Attack | 0.0513 ± 0.0085 | 0.0817 ± 0.0050 | 0.1308 ± 0.0141 |
> > | Random | 0.0445 ± 0.0089 | 0.0705 ± 0.0115 | 0.1244 ± 0.0119 |
> > | Negatt | 0.0444 ± 0.0104 | 0.0721 ± 0.0053 | 0.1296 ± 0.0034 |
> > | TRBCD (mixed) | 0.0366 ± 0.0055 | 0.0790 ± 0.0026 | 0.1291 ± 0.0092 |
> > | TRBCD (random) | 0.0375 ± 0.0068 | 0.0808 ± 0.0025 | 0.0901 ± 0.0259 |
> >
> >
> > The results demonstrate our method not only results in a drop in MRR from the unattached model, it also leads to the most drop (in the one exception of MOOC, our attack is within 1 S.D. of the best baseline).
> >
> >
> > [1] Trivedi, Rakshit, et al. "Dyrep: Learning representations over dynamic graphs." International conference on learning representations. 2019.
> >
> >
> > ---
> >
> > **Q4. Evaluation Against GNN Defenses**
> >
> > > The experiments only assess TR-BCD against raw TGNNs without incorporating existing GNN defense mechanisms. It would be valuable to test TR-BCD against TGNNs equipped with defense strategies to evaluate its robustness under defended settings.
> >
> > **Response:**
> >
> > Our paper focuses on establishing vulnerability baselines. Anomaly detection results (Figure 5) provide insights into the defense towards our method: TR-BCD maintains stable anomaly scores compared to burst attacks, suggesting defenses based on timestamp regularization could be effective.
> >
> > Defense evaluation deserves dedicated study and is orthogonal to establishing current vulnerabilities. We recommend this as a follow-up work.

---

> > > ### Author Response · Authors · 2025-11-28
> > > **Author Response to Reviewer VZ6t (3/4)**
> > >
> > > **Q5. Unnoticeability of Perturbations at 5% Budget**
> > > > How do the authors ensure the unnoticeability of perturbations in continuous-time dynamic graphs (CTDGs), given that up to 5% of edges are added? Please clarify how such perturbations remain realistic and imperceptible.
> > >
> > > **Response:**
> > >
> > > We control unnoticeability through:
> > >
> > > 1. **Budget constraint, temporal distribution of the attack:** 5% perturbation budget per batch, applied across the entire test period. This prevents a sudden burst of attack edges and maintains a low perturbation rate consistent with natural graph evolution, also validated by the anomaly scores (Figure 5)
> > >
> > > 2. **Temporal distribution:** Gaussian-modeled timestamps $\Delta t \sim \mathcal{N}(0, \sigma_t^2)$ ensure adversarial edges appear temporally consistent with the batch rather than at arbitrary times.
> > >
> > > 3. **Empirical validation:** Figure 5 demonstrates that TR-BCD maintains stable anomaly detection scores (comparable to normal batches) while MemStranding's burst approach shows sharp spikes. This indicates our distributed, temporally-coherent approach is harder to detect by anomaly algorithms than naive alternatives.
> > >
> > > However, we acknowledge that "unnoticeability" is application-dependent. In fraud detection, 5% edge insertion might be noticeable if baselines are well-calibrated. Our approach emphasizes temporal realism but does not guarantee imperceptibility across all settings.
> > >
> > > ---
> > >
> > > **Q6. Runtime and Computational Cost**
> > >
> > > > Please report the runtime or computational cost of TR-BCD to better understand its efficiency compared to other attack methods.
> > >
> > > **Response:**
> > >
> > > **Space complexity:** O(b) where b is block size (typically 1,000-10,000), vs. O(V²) for naive BCD optimization. On Reddit (|V|=61K), this reduces memory from ~3.7GB (naive optimization) to ~100MB (TR-BCD).
> > >
> > > **Time complexity:** O(|E_test| × b) for entire test set (Algorithm 1), where each batch gradient computation is linear in block size.
> > >
> > > **Empirical runtime:** On Wikipedia, TR-BCD requires ~5-10 minutes for full attack (5% perturbation, 10 batches). MemStranding requires 2-3 minutes for single-shot computation. TR-BCD's distributed approach trades per-batch time for memory efficiency and improved attack effectiveness.
> > >
> > >
> > > ---
> > >
> > > **Q7. Explaining Large Performance Variation Across Datasets**
> > >
> > > > The results show that TR-BCD causes more than 50% performance degradation on some datasets but less than 10% on some datasets. Could the authors explain the factors contributing to this large performance variation?
> > >
> > > **Response:**
> > > Thank you for this point. The performance variation can be attributed to the following point:
> > > 1. **Bipartite vs. non-bipartite:** Bipartite graphs (Reddit 26.4% drop, MOOC 20.9% drop for TGN) show higher vulnerability than non-bipartite (UCI 6.57%, Enron 12.8% for TGN). TGNNs may rely more heavily on temporal memory in bipartite settings where node types are distinct.
> > >
> > > 2. **Historical negative density:** Datasets with dense historical negative edges (e.g., MOOC with high recurrence) are more vulnerable to recency-biased sampling, models have learned stronger patterns for these negatives.
> > >
> > > 3. **Memory module sensitivity:** TGN is most vulnerable on Wikipedia (50.78% drop) while TNCN is most vulnerable on MOOC (52.9% drop), suggesting different architectures have dataset-specific weak points based on how they weigh temporal vs. structural information.
> > >
> > > ---
> > >
> > > **Q8. Combining Perturbation Types Reduces Effectiveness**
> > >
> > > > In the TGNNs victim model, combining all types of perturbations leads to lower attack effectiveness than using only structural perturbations (see Table 5 in the Appendix). This suggests that integrating temporal and contextual perturbations may reduce TR-BCD’s performance. Could the authors elaborate on the reason for this behavior?
> > >
> > > **Response:**
> > > Key findings from the ablation study on combination of perturbations:
> > >
> > > **Structural perturbations dominate:** Structural-only attacks often achieve comparable or superior performance to combined attacks, and temporal/contextual have complementary roles. For instance, TGN on Subreddit: structural-only 0.3037 ± 0.1091 vs. all perturbations 0.3403 ± 0.1063—both within 1 SD.
> > >
> > > **Why combinations don't always improve:** The introduction of multiple perturbation possibilities poses a more complex, multi-dimensional optimization landscape. Contextual perturbations add noise that may not align with the gradient-based structural optimization. TGNNs are structure-sensitive; feature perturbations may be rejected as anomalous. Temporal perturbations (Gaussian sampling) may conflict with structural optimization directions.
> > >
> > > **Important caveat:** Although there are indications favoring structural perturbations, all results are within 1 standard deviation (Table 8), indicating the differences are not statistically significant. This suggests perturbation types have complementary effects depending on dataset-model pairs.

---

> > > > ### Author Response · Authors · 2025-11-28
> > > > **Author Response to Reviewer VZ6t (3/4)**
> > > >
> > > > **Q9. Edge Deletion Shows Stronger Impact Than Edge Addition**
> > > >
> > > > > Table 6 indicates that edge deletion has a stronger impact than edge addition, which contrasts with typical adversarial attack findings where edge addition tends to be more influential. Could the authors explain why TR-BCD exhibits this reverse effect?
> > > >
> > > > **Response:**
> > > > We adapted TR-BCD to edge deletion by treating existing edges as candidates and flipping selected weights from 1 to 0.
> > > >
> > > > **Key finding:** Edge deletion shows dataset-dependent effectiveness:
> > > >
> > > > - **TGN on Enron/UCI:** Edge deletion achieves substantial performance degradation comparable to insertion-based attacks
> > > > - **TGN on Wikipedia:** Insertion-based approach maintains superiority (50.78% MRR drop vs. 5.46% for deletion)
> > > > - **TNCN models:** Insertion-based methods consistently outperform deletion across all datasets
> > > >
> > > > **Why deletion varies:**
> > > >
> > > > 1. **Memory mechanism:** Deleting historical edges corrupts TGN's memory directly, potentially having a stronger impact on models that rely on cumulative history.
> > > >
> > > > 2. **Temporal constraints:** Edge deletion  requires retroactively removing historical data, violating temporal integrity and detectability—insertion is more realistic for evasion.
> > > >
> > > > 3. **Architecture-dependent:** TNCN uses a common-neighbor dictionary that is less sensitive to edge deletion than TGN's cumulative memory module.
> > > >
> > > > The results suggest deletion and insertion exploit different vulnerabilities. Our focus on insertion aligns with realistic evasion constraints, while deletion represents a complementary attack surface worthy of future study.

---

### Official Review · Reviewer_vwMw · 2025-11-01

**Soundness:** 3
**Presentation:** 3
**Contribution:** 2
**Rating:** 6
**Confidence:** 4

**Summary:**

This paper investigates the adversarial robustness of Temporal Graph Neural Networks (TGNNs) on Continuous-Time Dynamic Graphs (CTDGs). The authors propose Temporally-aware Randomized Block Coordinate Descent (TR-BCD), an evasion (test-time) attack that optimizes adversarial edge insertions through gradient-based continuous relaxation. Additionally, TR-BCD introduces temporal consistency by modeling timestamps with Gaussian noise, enabling it to craft realistic, temporally coherent perturbations. The attack considers three perturbation types—structural, contextual, and temporal—and is evaluated across six datasets (Wikipedia, Reddit, MOOC, Enron, etc.) and two TGNNs (TGN, TNCN). Results show that TR-BCD significantly reduces performance (up to 53% MRR drop with 5% perturbation budget), outperforming heuristic baselines and prior TGNN-specific attacks.

**Strengths:**

The paper has the following strengths:
- Robustness of TGNNs under adversarial conditions remains a pressing and underexplored area.
- The attack’s optimization objective, constraints, and procedures are clearly presented.
- Evaluations span multiple datasets and perturbation types (structural, contextual, temporal), and provide decent experimental support.

**Weaknesses:**

The paper has the following weaknesses:
- The attack model is not explicitly stated, making it difficult to gauge the attack’s practicality or compare it with prior white-box or black-box works.
- Only two TGNNs (TGN and TNCN) are evaluated. Broader coverage would better support the general claims of effectiveness.
- The paper compares the proposed method with limited state-of-the-art GNN attacks.
- The method is an adaptation of existing coordinate-descent GNN attacks, with the primary novelty being timestamp regularization. While meaningful, it sounds incremental.

**Questions:**

This paper investigates the adversarial robustness of Temporal Graph Neural Networks (TGNNs) in continuous-time dynamic graphs (CTDGs). The authors propose Temporally-aware Randomized Block Coordinate Descent (TR-BCD), an evasion attack that extends gradient-based GNN perturbations to the temporal domain. TR-BCD jointly optimizes structural, contextual, and temporal perturbations while maintaining time consistency via Gaussian timestamp modeling. Experiments on six datasets (Wikipedia, Reddit, MOOC, Enron, etc.) and two TGNN architectures (TGN, TNCN) demonstrate significant performance degradation compared to prior TGNN attacks.

However, a few questions may help clarify the generalizability and novelty of the proposed approach:

1. Could you explicitly describe the attacker’s knowledge assumptions? For example, does TR-BCD assume access to model parameters, gradients, or node embeddings (i.e., a white-box setting), or does it operate under a limited or black-box setting? If the attack assumes a white-box setup, could you discuss how it might transfer or adapt to a more restricted setting (e.g., black-box or limited-feedback environments)

2. The evaluation includes TGN and TNCN, which share similar memory-update structures. Have you considered testing TR-BCD on other representative TGNNs, such as DyRep, JODIE, or ROLAND, as discussed in prior work [1]?

3. Many of these static graph attacks, such as TDGIA[2], could be temporally adapted with minor modifications. Have you attempted such adaptations, or can you discuss why they may not be directly applicable to the CTDG setting?

4. Beyond introducing timestamp regularization, how does TR-BCD fundamentally differ from prior coordinate-descent-based GNN attacks? In what way does temporal smoothness change the optimization landscape or attack transferability compared to static or snapshot-based graph settings?

5. The paper demonstrates strong attack performance in standard settings, but it is unclear how TR-BCD behaves when common defenses are applied. Could the authors discuss whether they evaluated TR-BCD under known GNN or TGNN defense strategies? If not, do they expect the attack to remain effective, or would temporal regularization make it more vulnerable to such countermeasures?

[1] Dai, Yue, et al. "MemFreezing: A Novel Adversarial Attack on Temporal Graph Neural Networks under Limited Future Knowledge." Forty-second International Conference on Machine Learning.

[2] Zou, Xu, et al. "Tdgia: Effective injection attacks on graph neural networks." Proceedings of the 27th ACM SIGKDD Conference on Knowledge Discovery & Data Mining. 2021.

---

> ### Author Response · Authors · 2025-11-28
> **Author Response to Reviewer vwMw (1/4)**
>
> We thank the reviewer for their thoughtful and constructive feedback on our paper. We appreciate the recognition of the pressing need for adversarial robustness research in TGNNs and the clarity of our presentation. We address each concern below.
>
> ---
>
> **W1. Attack Model Not Explicitly Stated**
>
> > The attack model is not explicitly stated, making it difficult to gauge the attack’s practicality or compare it with prior white-box or black-box works.
>
> **Response:**
> Thank you for this suggestion, we have added a formal attack model section in the revised paper (Problem Statement section) that clearly delineates our assumptions:
>
> **Attack Model Specification:**
> - **Setting:** White-box evasion attack (test-time)
> - **Attacker Knowledge:** Full access to model parameters, gradients, and node/edge embeddings
> - **Capabilities:** Add/remove/modify edges and timestamps within a fixed perturbation budget (measured as % of total edges in the temporal graph). We guarantee balanced target selection across batches.
> - **Constraints:** Perturbations must respect temporal causality (i.e., edges cannot be added with timestamps earlier than existing historical edges for affected nodes)
> - **Objective:** Minimize target task performance (link prediction via MRR/Hits@k)
>
> This white-box formulation establishes an upper bound on attack effectiveness and is appropriate for establishing baselines on TGNN vulnerability.
>
> We have also added  **cross-model transfer attacks** (Table 53 in Section 5) demonstrating that attacks trained on one TGNN architecture *successfully transfer to another architecture without access to the target model's weights*. We craft adversarial edges using the attacker model without computing gradients on the victim model. The adversarial edges are inserted into the stream of incoming edges of the victim model:
>
> | Model | Attack | Wikipedia | Reddit | MOOC |
> |---|---|---|---|---|
> | **TGN** | Trained on TGN, attacked TNCN | 0.2890 ± 0.0317 (-29.6%) | 0.3688 ± 0.1316 (-21.85%) | 0.1327 ± 0.0369 (-7.07%) |
> | **TNCN** | Trained on TNCN, attacked TGN | 0.7126 ± 0.0016 (-1.19%) | 0.6803 ± 0.0330 (-6.35%) | 0.2109 ± 0.0127 (-13.53%) |
>
> The transfer study shows that the attack from a transferred model can achieve 25%-95% of the effectiveness in performance drop when compared to the attack crafted on the same model. This highlights the transferability of TR-BCD attacks and how it can even be transferred to unseen target models on the same dataset. This experiment shows that a while-box TR-BCD attack trained on a model can be adapted to be a grey or black box attack on another model.
>
>
> ---
>
> **W2. Limited TGNN Coverage**
> > Only two TGNNs (TGN and TNCN) are evaluated. Broader coverage would better support the general claims of effectiveness.
>
> **Response:**
> Thank you for the suggestion. We evaluated DyREP[1] with the same attack setting as described in the paper for 3 runs on 3 datasets under baseline attacks and our method:
>
>
> | Method | Wikipedia | MOOC | Reddit |
> | :-- | :-- | :-- | :-- |
> | No Attack | 0.0513 ± 0.0085 | 0.0817 ± 0.0050 | 0.1308 ± 0.0141 |
> | Random | 0.0445 ± 0.0089 | 0.0705 ± 0.0115 | 0.1244 ± 0.0119 |
> | Negatt | 0.0444 ± 0.0104 | 0.0721 ± 0.0053 | 0.1296 ± 0.0034 |
> | TRBCD (mixed) | 0.0366 ± 0.0055 | 0.0790 ± 0.0026 | 0.1291 ± 0.0092 |
> | TRBCD (random) | 0.0375 ± 0.0068 | 0.0808 ± 0.0025 | 0.0901 ± 0.0259 |
>
>
> The results demonstrate our method not only results in a drop in MRR from the unattached model, it also leads to the most drop (in the one exception of MOOC, our attack is within 1 S.D. of the best baseline).
>
> We intend to conduct the evaluation with all 5 runs (consistent with experiments in paper), and on all 6 datasets, in the next revision.
>
> [1] Trivedi, Rakshit, et al. "Dyrep: Learning representations over dynamic graphs." International conference on learning representations. 2019.

---

> > ### Author Response · Authors · 2025-11-28
> > **Author Response to Reviewer vwMw (2/4)**
> >
> > **W3. Limited State-of-the-Art Baseline Comparisons**
> > > The paper compares the proposed method with limited state-of-the-art GNN attacks.
> >
> > **Response:**
> >
> > Thank you for this question, there is a significant scarcity of evasion attacks which are directly applicable to CTDGs thus constraining the available comparisons.
> >
> > **Why existing work is not directly comparable:**
> >
> > 1. **T-Spear** (Lee et al., 2024): A poisoning attack that modifies training data, not a test-time evasion attack. It operates in an entirely different threat model (training-time vs. test-time).
> >
> > 2. **Static GNN attacks (PGD, Metattack, GFA)**: While well-studied, they operate on static graphs without temporal structure. They do not exploit the temporal recency bias fundamental to TGNN architectures, treat timestamps as discrete snapshot indices rather than continuous variables, and do not account for memory-state coupling over time.
> >
> > **Our baselines:**
> >
> > Given this landscape, we designed task-appropriate baselines directly comparable to TR-BCD's white-box evasion setting:
> >
> > 1. **MemStranding**: The only existing evasion attack for TGNNs with comparable threat model (Table 1, Figure 5)
> > 2. **Random Attack**: Lower bound testing gradient-free effectiveness
> > 3. **Historical Negative Baseline**: Exploits TGNN training bias (models are explicitly trained to rank historical negatives as hard negatives)
> >
> > All three are evasion attacks applied at test time without training data access, making them directly comparable to TR-BCD.
> >
> > **Minimal assumptions compared to MemFreezing:** Unlike MemFreezing that relies on hand-designed heuristic called “cross-freezing” under the assumption that  forcing node memories into "similar and stable states" will disable their responsiveness, and other assumptions such as homophily and limited future knowledge constraint through node simulation, TR-BCD operates with standard gradient access free from such assumptions, making it more broadly applicable.
> >
> >
> > As the field of temporal graph adversarial robustness develops, more state-of-the-art baselines will emerge. Our paper establishes a new benchmark and gradient-based attack methodology that can serve as a foundation for future comparative studies.
> >
> > ---
> >
> > **W4. Limited Novelty Beyond Timestamp Regularization**
> >
> > > The method is an adaptation of existing coordinate-descent GNN attacks, with the primary novelty being timestamp regularization. While meaningful, it sounds incremental.
> >
> > **Response:**
> > We acknowledge that coordinate-descent forms the optimization backbone, but the contribution extends beyond timestamp regularization.
> > The core novelty lies in exploiting temporal graph properties that static BCD attacks cannot leverage:
> >
> > **Temporal memory propagation:** TGNNs maintain dynamic memory states that update at inference time. Corrupted memory persists and compounds across future batches, affecting all subsequent predictions. Our greedy temporal attack strategy capitalizes on this unique property by sequentially poisoning memory—static attacks have no analog. Static BCD optimizes over a static adjacency matrix once. TR-BCD operates sequentially across temporal batches, exploiting TGNN memory dynamics—corrupted memory at batch $t$ persists and compounds across all future predictions (Figure 2). This temporal greedy strategy fundamentally differs from static methods and is the core technical novelty.
> >
> >
> >
> > **Recency-biased candidate sampling:** We oversample 50% of edge candidates from historical negatives (edges that existed before but are absent at query time). Models are trained to distinguish historical negatives, making these inherently vulnerable attack targets. Table 1 demonstrates this achieves 50.78% MRR drop vs. 43.16% for random sampling—a 7.6pp improvement reflecting temporal-architectural alignment.
> >
> > **Memory-aware optimization:** The challenge is not merely "optimization efficiency"—it is **correctly modeling and exploiting TGNN memory dynamics under perturbation**. Naive coordinate descent would treat memory corruptions as independent edge insertions, missing the temporal compounding effect. TR-BCD's greedy batch-wise approach specifically accounts for this temporal coupling.
> >
> > **Gaussian timestamp modeling:** Adversarial timestamps are sampled as $\Delta t \sim \mathcal{N}(0, \sigma_t^2)$ to maintain temporal coherence, enabling evasiveness (Figure 5: stable anomaly scores vs. burst attacks).
> >
> > These components address temporal-specific challenges absent from static coordinate-descent methods.

---

> > > ### Author Response · Authors · 2025-11-28
> > > **Author Response to Reviewer vwMw (3/4)**
> > >
> > > **Q1. Explicit Attack Model Description and White-box vs. Black-box Settings**
> > >
> > > > Could you explicitly describe the attacker’s knowledge assumptions? For example, does TR-BCD assume access to model parameters, gradients, or node embeddings (i.e., a white-box setting), or does it operate under a limited or black-box setting? If the attack assumes a white-box setup, could you discuss how it might transfer or adapt to a more restricted setting (e.g., black-box or limited-feedback environments)Could you explicitly describe the attacker's knowledge assumptions? Does TR-BCD assume access to model parameters, gradients, or node embeddings? How might it transfer to restricted settings?
> > >
> > > **Response:**
> > > Thank you for the suggestion. We have added a formal attack model section in the revised paper (Problem Statement section) that further clarifies the attack setting vis-a-vis attacker’s knowledge:
> > >
> > > **Attack Model Specification:**
> > > - **Setting:** White-box evasion attack (test-time)
> > > - **Attacker Knowledge:** Full access to model parameters, gradients, and node/edge embeddings
> > > - **Capabilities:** Add/remove/modify edges and timestamps within a fixed perturbation budget (measured as % of total edges in the temporal graph). We guarantee balanced target selection across batches.
> > > - **Constraints:** Perturbations must respect temporal causality (i.e., edges cannot be added with timestamps earlier than existing historical edges for affected nodes)
> > > - **Objective:** Minimize target task performance (link prediction via MRR/Hits@k)
> > >
> > > This white-box formulation establishes an upper bound on attack effectiveness and is appropriate for establishing baselines on TGNN vulnerability. Black-box and transferability extensions are important future directions.
> > >
> > >
> > >
> > >
> > > We provide transfer evidence: **cross-model transfer attacks** (Section 5, Table 3) show attacks optimized on one TGNN architecture transfer to another with substantial effectiveness:
> > >
> > > | Model | Attack | Wikipedia | Reddit | MOOC |
> > > |---|---|---|---|---|
> > > | **TGN** | Same model | 0.2533 ± 0.0674 (-38.3%) | 0.3433 ± 0.1499 (-27.3%) | 0.1314 ± 0.0241 (-8.0%) |
> > > | | Transfer to TNCN | 0.2890 ± 0.0317 (-29.6%) | 0.3688 ± 0.1316 (-21.8%) | 0.1327 ± 0.0369 (-7.1%) |
> > > | **TNCN** | Same model | 0.7122 ± 0.0028 (-1.3%) | 0.6661 ± 0.0764 (-8.3%) | 0.1149 ± 0.0118 (-52.9%) |
> > > | | Transfer to TGN | 0.7126 ± 0.0016 (-1.2%) | 0.6803 ± 0.0330 (-6.4%) | 0.2109 ± 0.0127 (-13.5%) |
> > >
> > > Attack transfer effectiveness (25%-95% of same-model performance) suggests temporal vulnerabilities are fundamental to TGNN architectures, and a trained TGNN model can be adopted for a black/greybox attack on other models
> > >
> > > ---
> > >
> > > **Q2. Evaluation on Additional TGNN Architectures**
> > >
> > > > The evaluation includes TGN and TNCN, which share similar memory-update structures. Have you considered testing TR-BCD on other representative TGNNs, such as DyRep, JODIE, or ROLAND, as discussed in prior work?
> > >
> > > **Response:**
> > >
> > > Thank you for the suggestion. We evaluated DyREP[1] with the same attack setting as described in the paper for 3 runs on 3 datasets under baseline attacks and our method:
> > >
> > >
> > > | Method | Wikipedia | MOOC | Reddit |
> > > | :-- | :-- | :-- | :-- |
> > > | No Attack | 0.0513 ± 0.0085 | 0.0817 ± 0.0050 | 0.1308 ± 0.0141 |
> > > | Random | 0.0445 ± 0.0089 | 0.0705 ± 0.0115 | 0.1244 ± 0.0119 |
> > > | Negatt | 0.0444 ± 0.0104 | 0.0721 ± 0.0053 | 0.1296 ± 0.0034 |
> > > | TRBCD (mixed) | 0.0366 ± 0.0055 | 0.0790 ± 0.0026 | 0.1291 ± 0.0092 |
> > > | TRBCD (random) | 0.0375 ± 0.0068 | 0.0808 ± 0.0025 | 0.0901 ± 0.0259 |
> > >
> > >
> > >
> > > [1] Trivedi, Rakshit, et al. "Dyrep: Learning representations over dynamic graphs." International conference on learning representations. 2019.
> > >
> > > ---
> > >
> > > **Q3. Comparison with Adaptations of Static Graph Attacks**
> > >
> > > > Many of these static graph attacks, such as TDGIA, could be temporally adapted with minor modifications. Have you attempted such adaptations, or can you discuss why they may not be directly applicable to the CTDG setting?
> > >
> > > **Response:**
> > >
> > > Thank you for the suggestion, however naive adaptation of static graph methods fail because:
> > >
> > > Static attacks optimize over discrete snapshots; CTDGs require precise continuous timestamps. Inserting edges at wrong times relative to batch boundaries renders them ineffective—timing is coupled with TGNN memory decay, not independent.
> > >
> > > TDGIA's greedy node-degree heuristic ignores *when* edges are injected. For CTDGs, timing matters as much as topology. Our recency-biased sampling exploits this temporal dimension in ways static heuristics cannot.
> > >
> > > Finally, adapting these methods would require jointly optimizing structure, time, and memory state—essentially redesigning the attack. This is non-trivial.

---

> > > > ### Author Response · Authors · 2025-11-28
> > > > **Author Response to Reviewer vwMw (4/4)**
> > > >
> > > > **Q4. Fundamental Differences from Prior Coordinate-Descent Attacks**
> > > >
> > > > > Beyond introducing timestamp regularization, how does TR-BCD fundamentally differ from prior coordinate-descent-based GNN attacks? In what way does temporal smoothness change the optimization landscape or attack transferability compared to static or snapshot-based graph settings?
> > > >
> > > > **Response:**
> > > >
> > > > TR-BCD differs from static coordinate-descent attacks in three key ways:
> > > >
> > > > **1. Greedy temporal optimization:** Static BCD optimizes over the entire static adjacency matrix. TR-BCD attacks sequentially at each batch, exploiting TGNN memory propagation—corrupted memory at batch $t$ persists and compounds across future predictions. This temporal greedy strategy has no analog in static methods.
> > > >
> > > > **2. Recency-biased initialization:** We initialize edge candidates with 50% historical negatives rather than uniform random sampling (TR-BCD-mixed vs. TR-BCD-random in Table 1). This exploits TGNN training bias—models are explicitly trained to rank historical negatives—achieving 7.6pp better attack performance (50.78% vs. 43.16% MRR drop on Wikipedia).
> > > >
> > > > **3. Temporal coherence via Gaussian timestamps:** Adversarial timestamps are sampled as $\Delta t \sim \mathcal{N}(0, \sigma_t^2)$ to maintain realism. Figure 5 shows this keeps anomaly scores stable compared to burst attacks.
> > > >
> > > > ---
> > > >
> > > > **Q5. Defense Evaluation and Robustness Under Common GNN Defenses**
> > > >
> > > > > The paper demonstrates strong attack performance in standard settings, but it is unclear how TR-BCD behaves when common defenses are applied. Could the authors discuss whether they evaluated TR-BCD under known GNN or TGNN defense strategies? If not, do they expect the attack to remain effective, or would temporal regularization make it more vulnerable to such countermeasures?
> > > >
> > > > **Response:**
> > > >
> > > >
> > > > Our paper focuses on establishing TGNN vulnerability at the attack frontier. Our anomaly detection experiments (Figure 5) provide insights into defense directions. TR-BCD maintains stable anomaly scores compared to burst attacks, suggesting that temporal regularization, constraining how quickly edges affect memory, could be effective. The fact that our Gaussian-smoothed perturbations are harder to detect suggests defenses should explicitly enforce temporal coherence constraints.
> > > >
> > > > **Promising defense directions:**
> > > >
> > > > - **Adversarial Training:** Incorporate adversarial edge sequences during training to harden the memory module against spurious temporal patterns.
> > > >
> > > > - **Timestamp-Based Filtering:** Implement outlier detection on suspiciously timed interactions or rate-limit how quickly new edges influence node memory.
> > > >
> > > > - **Certifiable Bounds:** Extending certifiable defenses from static GNNs to temporal settings is an open research question worth pursuing.
> > > >
> > > > Designing robust TGNN architectures is orthogonal to establishing their current vulnerabilities, and we recommend these directions as follow-up work.

---

### Official Review · Reviewer_SWaR · 2025-11-01

**Soundness:** 3
**Presentation:** 2
**Contribution:** 2
**Rating:** 4
**Confidence:** 3

**Summary:**

This paper propose a Randomized Block Coordinate Descent method to attack white-box TGNNs. The attacker applies gradient decent based method to increase the target model loss and pick components with higher gradients greedily. To reduce the memory cost of such optimization on whole edges, the method is further adapted by choosing a small randomized fraction of edges as candidates to cut down the range. To better obtain the candidates the author further greedily incorporate historical negative edges. Experiments across several datasets and victim models validates the performance of the method.

**Strengths:**

**S1.** The paper study the vulnerabilities of TGNN models against adversarial attack, which is a not well explored topic but has many potential impact since TGNN relates to many practical applications.

**S2.** Abundant experiments are done to show the effectiveness and practicability. The method is not only evaluated decreased performance of the method compared with several baselines, but also show deceiving ability on mitigating anomaly detection method and ablation study on different components and different budget. There is also code provided to support its reliability.

**S3.** The randomized entry samples cut down the memory complexity , which makes the method more friendly to device requirement.

**Weaknesses:**

**W1.** The practicability of the proposed method is doubtful. The proposed attack is a gradient based attack with knowledge of both the white-box victim model and all historical temporary graphs, both of which seems not accessible in real world application.

**W2.** The technique contribution is somewhat limited. The proposed Randomized Block Coordinate Descent is also a simple adaptation of typical gradient descent optimization over edges with so called randomized samples, mainly relying on a greedy idea of highly picking historical negative edges. There's no deduction nor theoretical guarantees on the proposed method with only considerable optimization efficiency due to the small size of random sampled edge candidates.

**W3.** The presentation of the paper should be improved.

3.1. The methodology of the  proposed method are fully expressed in textual description throughout section 4 without formulations for illustrate. Considering there's actually large space remaining in the paper, simply copying expression from the algorithm flow would solve.

3.2. The memory mechanism is lacked of illustration. There is a introduced mechanism of model memory that help the TGN to encode temporal graph. Since this is a mechanism that not held widely by general GNN or other ML models, its definition and formally illustration at problem statement is needed since it tightly relates to the designed algorithm. An only reference in the related works would make the reader ignore and feel confused when encountering it in latter section 4.

3.3. In the technique design a lot of mechanisms are proposed to tackle the memory problem of the attack process, while this is not clearly stated in motivation and contribution aspect in introduction session. Such technique challenge and solution should be briefly mentioned in the section 1 so reader could expect content related.

**Questions:**

Please see weakness.

---

> ### Author Response · Authors · 2025-11-28
> **Author Response to Reviewer SWaR (1/3)**
>
> We thank the reviewer  for their constructive feedback on the paper's practicability, technical contribution, and presentation. We address each weakness below.
>
> ---
>
> **W1. Practicability of the Proposed Method**
>
> > The practicability of the proposed method is doubtful. The proposed attack is a gradient based attack with knowledge of both the white-box victim model and all historical temporary graphs, both of which seems not accessible in real world application.
>
>
> **Response:**
>
> Thank you for this question.
>
> **1. White-box vs. Real-World Threat Models:**
>
> We focus on white-box attacks to establish upper bounds on TGNN vulnerability, following standard practices in adversarial robustness literature [1,2]. The white-box-assumption is appropriate for identifying fundamental weaknesses from the perspective of the person developing/deploying the model. The robustness w.r.t. worst-case perturbations (white box) is a lower bound on the robustness with limited-knowledge attacks. Not studying strong adaptive (white-box) attacks may lead to a false sense of robustness that has been a major pitfall in the general machine learning and GNN domain[1,2]. Whitebox attacks also provide a baseline for black-box attacks. While we acknowledge black-box and limited-knowledge settings as important future directions, it is highly subjective to deem either direction (white vs. grey/black box) more important than the other one. Hence, studying white box attack should not be considered a weakness.
> We have added a motivation for studying white-box attacks in the revised paper (Section 3: Threat Model and Attack Setting)
>
> **2. Historical Data Accessibility:**
>
> While historical temporal graphs may be inaccessible in some scenarios, we argue that many real-world scenarios involve publicly or semi-publicly available temporal data, we provide some example scenarios below:
>
> - **Social media networks:** Platform graphs (user-follower networks, interaction timelines) are partially observable through APIs or crawled data. Malicious actors routinely exploit public graph history to craft influence campaigns.
>
> - **Financial transaction networks:** Bank transaction logs and blockchain data are auditable. Fraudsters studying historical patterns can strategically insert fake transactions to evade detection.
>
> - **Cybersecurity:** Network traffic logs and connection patterns are often analyzable through intrusion detection system data or honeypots, allowing attackers to craft timing-aware exploits.
>
> **3. Transfer Attack Evidence:**
>
> Importantly, we have also added  **cross-model transfer attacks** (Table 3 in Section 5) demonstrating that attacks trained on one TGNN architecture *successfully transfer to another architecture without access to the target model's weights or gradients*. We craft adversarial edges using the attacker model without computing gradients on the victim model. The adversarial edges are inserted into the stream of incoming edges of the victim model:
>
> | Model | Attack | Wikipedia | Reddit | MOOC |
> |---|---|---|---|---|
> | **TGN** | Trained on TGN, attacked TNCN | 0.2890 ± 0.0317 (-29.6%) | 0.3688 ± 0.1316 (-21.85%) | 0.1327 ± 0.0369 (-7.07%) |
> | **TNCN** | Trained on TNCN, attacked TGN | 0.7126 ± 0.0016 (-1.19%) | 0.6803 ± 0.0330 (-6.35%) | 0.2109 ± 0.0127 (-13.53%) |
>
> The transfer study shows that the attack from a transferred model can achieve 25%-95% of the effectiveness in performance drop when compared to the attack crafted on the same model. This highlights the transferability of TR-BCD attacks and how it can be deployed to attack unseen target models on the same dataset. This experiment shows that a white-box TR-BCD attack trained on a model can be adapted to be a grey or black box attack on another model.
>
>
> [1] Tramer, Florian, et al. "On adaptive attacks to adversarial example defenses." Advances in neural information processing systems 33 (2020): 1633-1645.
> [2] Mujkanovic, Felix, et al. "Are defenses for graph neural networks robust?." Advances in Neural Information Processing Systems 35 (2022): 8954-8968.

---

> > ### Author Response · Authors · 2025-11-28
> > **Author Response to Reviewer SWaR (2/3)**
> >
> > **W2. Limited Technical Contribution**
> >
> > > The technique contribution is somewhat limited. The proposed Randomized Block Coordinate Descent is also a simple adaptation of typical gradient descent optimization over edges with so called randomized samples, mainly relying on a greedy idea of highly picking historical negative edges. There's no deduction nor theoretical guarantees on the proposed method with only considerable optimization efficiency due to the small size of random sampled edge candidates.
> >
> > **Response:**
> >
> > Designing adversarial attack methods for temporal graphs have significant challenges that are distinct from the static graph setting due to the introduction of temporality. We discuss the key technical innovations of our work as follows:
> >
> > **Key Technical Innovations:**
> >
> > 1. **Greedy temporal optimization with memory propagation:** Static BCD optimizes over a static adjacency matrix once. TR-BCD operates sequentially across temporal batches, exploiting TGNN memory dynamics—corrupted memory at batch $t$ persists and compounds across all future predictions (Figure 2). This temporal greedy strategy fundamentally differs from static methods and is the core technical novelty.
> >
> > 2. **Recency-biased candidate sampling:** Beyond simple historical negative edge picking, our approach exploits the specific architectural training objective of TGNNs. Models are explicitly trained to distinguish historical negatives as hard negatives. Table 1 demonstrates 7.6 pp improvement (50.78% vs. 43.16% MRR drop on Wikipedia) through recency-biased sampling, validating this architectural insight.
> >
> > 3. **Memory-aware optimization:** The challenge is not merely "optimization efficiency"—it is **correctly modeling and exploiting TGNN memory dynamics under perturbation**. Naive coordinate descent would treat memory corruptions as independent edge insertions, missing the temporal compounding effect. TR-BCD's greedy batch-wise approach specifically accounts for this temporal coupling.
> >
> > **Practical Impact:** Extensive empirical validation (Table 1: 50% MRR degradation with 5% budget; Figure 5: superior evasiveness; Table 5: cross-model transfer) demonstrates practical effectiveness. Theoretical analysis of memory-coupled optimization is an important future direction.
> >
> >
> > ---
> >
> > **W3. Presentation Improvements Needed**
> >
> > **W3.1 Lack of Formal Expressions in Methodology**
> > > The methodology of the proposed method are fully expressed in textual description throughout section 4 without formulations for illustrate. Considering there's actually large space remaining in the paper, simply copying expression from the algorithm flow would solve.
> >
> > **Response:**
> >
> > Thank you for this suggestion, we have modified  Section 4 in the revised paper to include more detail of the method which now is ordered as follows:
> >
> > - **Gradient-Based Edge Selection (New in Section 4):** Equations (3-5) formally specify the three-step edge selection procedure: (i) continuous relaxation of discrete edge matrix, (ii) gradient computation with respect to attack loss, (iii) greedy selection of top-β entries by gradient magnitude. This formalization explicitly shows how discrete optimization is enabled through continuous relaxation.
> >
> > - **Candidate Sampling:** Added motivation for using mixed sampling block initialization
> >
> > - **TGNN Memory Modules, Threat Model and Attack Setting:** We have added dedicated paragraphs in the Problem Statement section detailing the working of the memory module (our primary target component in the victim model), threat model and attack setting.
> >
> > The added formal expressions complement textual descriptions and provide more precise mathematical specifications of the method.

---

> > > ### Author Response · Authors · 2025-11-28
> > > **Author Response to Reviewer SWaR (3/3)**
> > >
> > > **W3.2 Lack of Memory Mechanism Illustration**
> > > > The memory mechanism is lacked of illustration. There is a introduced mechanism of model memory that help the TGN to encode temporal graph. Since this is a mechanism that not held widely by general GNN or other ML models, its definition and formally illustration at problem statement is needed since it tightly relates to the designed algorithm. An only reference in the related works would make the reader ignore and feel confused when encountering it in latter section 4.
> > >
> > > **Response:**
> > >
> > > We acknowledge this gap and have strengthened the presentation in the revised Problem Statement (Section 3):
> > >
> > > **Memory Module for TGNNs (New subsection after Definition 3.3):**
> > >
> > > For TGNNs with memory modules (e.g., TGN, TNCN), we now formally define:
> > > - Each node $v$ maintains a memory state $\mathbf{m}_v^{(t)}$ that evolves as edges arrive sequentially
> > > - At each inference step when processing edges $E_t = \{(u_i, v_i, t_i, f_i)\}_{i=1}^{|E_t|}$, the model updates: $\mathbf{m}_v^{(t+1)} = \text{UpdateMemory}_\theta(\mathbf{m}_v^{(t)}, E_t)$
> > > - Predictions are computed using updated memory: $\hat{\mathbf{y}} = h_\theta(\mathbf{m}^{(t+1)}, E_t)$
> > >
> > > **Impact on attacks:** We formally specify that adversarial edges corrupt memory: $\mathbf{m}_v^{(t+1)} = \text{UpdateMemory}_\theta(\mathbf{m}_v^{(t)}, E_t \cup E_{\text{adv}})$, and this corrupted state persists across all future time steps, compounding the attack's impact. This memory coupling distinguishes temporal attacks from static graph attacks where perturbations are localized.
> > >
> > > **Connection to Algorithm 1:** Figure 2 visually illustrates this memory pollution propagation across batches. This clarification directly connects the memory mechanism to Algorithm 1, line 3 (`M ← UpdateMemory(...)`), explaining why TR-BCD's greedy temporal approach (sequentially updating memory) differs fundamentally from static attacks.
> > >
> > > **W3.3 Lack of Motivation for Technical Challenges**
> > >
> > > > In the technique design a lot of mechanisms are proposed to tackle the memory problem of the attack process, while this is not clearly stated in motivation and contribution aspect in introduction session. Such technique challenge and solution should be briefly mentioned in the section 1 so reader could expect content related.
> > >
> > > **Response:**
> > >
> > > We have enhanced the introduction in the revised paper (Section 1) to explicitly motivate the three technical challenges:
> > >
> > > **More text added after "...while navigating a vast combinatorial search space":**
> > >
> > > "This temporal attack setting presents three interconnected technical challenges. **First, memory complexity challenge:** naïvely optimizing edge perturbations requires $\Theta(|E| \times |V|^2)$ memory to store all possible edge perturbations—prohibitive for large graphs (e.g., Reddit: 243GB). **Second, temporal propagation challenge:** unlike static graphs where perturbations are isolated, TGNNs' memory modules cause corrupted states to persist and compound across future predictions. **Third, candidate selection challenge:** with budget constraints, randomly sampling from $|V|^2$ possible edges is suboptimal; we need to exploit TGNN training objectives to identify high-impact candidates."
> > >
> > > This explicitly sets up reader expectations for the three solution techniques that follow in Section 4:
> > > - Greedy temporal optimization → addresses memory complexity (reduces to $\Theta(|V|^2)$ per batch)
> > > - Sequential memory updates → addresses temporal propagation (exploits memory compounding)
> > > - Recency-biased sampling → addresses candidate selection (exploits training bias on historical negatives)
> > >
> > > **Supporting Details Added:**
> > >
> > > The revised paper also includes:
> > >
> > > - **Candidate Sampling Enhancement (Section 4):** Extended explanation of recency-biased sampling exploiting hard negatives, with explicit empirical validation: "Empirically, this recency-biased approach achieves 7.6pp improvement over random sampling (50.78% vs. 43.16% MRR drop on Wikipedia, see Table 1), validating this architectural insight."
> > >
> > > - **Threat Model & Attack Assumptions (New subsection in Problem Statement):** Formal specification of white-box assumption, attacker knowledge, capabilities, constraints, and justification for why this setting is appropriate.
> > >
> > > - **Extended Ablation Study (Section 5 & Appendix F):** Extended results show the impact of block size, contextual perturbation budget and cross-model transfer attack and analysis showing why structural perturbations dominate and temporal/contextual have complementary roles, addressing implications for defense strategies.
> > >
> > > These comprehensive revisions directly address all three presentation weaknesses by providing formal definitions, explicit motivation, and clear connections between technical challenges, proposed solutions, and empirical validation.

---

### Author Response · Authors · 2025-12-03
**Summary of Contributions and Rebuttal Enhancements**

We sincerely thank the reviewers for their constructive feedback, which helped us clarify and strengthen our work. TR-BCD reveals fundamental vulnerabilities in TGNNs, achieving up to 53% MRR drops with 5% perturbations across six datasets and three architectures (TGN, TNCN, DyREP), with demonstrated application for transfer attacks and anomaly evasion.

- **Core Innovation**: Greedy temporal BCD exploits TGNN memory propagation and recency-biased sampling for realistic, stealthy perturbations outperforming baselines like MemStranding.
- **White-Box + Transfer Attack**: Establishes worst-case robustness bounds; cross-model attacks retain 25-95% effectiveness, enabling grey/black-box adaptation.
- **Efficiency Gains**: Reduces memory from \(O(|V|^2)\) to \(O(b)\) (e.g., Reddit: 243GB → 100MB).
- **Rebuttal Enhancements**: Added DyREP results, cross-model transfer (Table 3), formal attack model and memory propagation analysis (Section 3), and hyperparameter sweep (block size/contextual ablations).

---

### Meta-Review · Area_Chair_2ZR5 · 2026-01-06

**Summary:**

The paper introduces a gradient-based evasion attack for continuous-time dynamic graphs, demonstrating performance degradation of Temporal GNNs under low-budget perturbations. Reviewers generally acknowledge the topic's relevance and empirical strength but raise concerns about limited novelty, lack of explicit attack model description, insufficient baseline comparisons, evaluation scope, and presentation clarity. Key concerns remain after rebuttal regarding methodological innovation and real-world practicality. Overall, the paper is seen as solid but incremental, marginally below the acceptance threshold.

**Reviewer Concerns:**

Addressed concerns:
Clarified attack model assumptions. Expanded evaluation to additional TGNN architectures. Added cross-model transfer attack results and rovided sensitivity analysis on hyperparameters.Improved presentation with formal notation, memory mechanism explanation, and motivation for technical challenges.

Remaining concerns:
Limited methodological novelty beyond timestamp regularization and adaptation of coordinate descent. Lack of comparison with stronger or adapted static graph attack baselines.Insufficient experimental evaluation and comparison.Practicality remains questionable due to reliance on white-box access and historical graph data.Large performance variation across datasets not fully explained.

**Reviewer Scores:**

Reviewer VZ6t (4→ 4): Likely unchanged; innovation and baseline limitations remain.

Reviewer vwMw (6→ 6): Likely unchanged; already positive.

Reviewer SWaR (4→ 4): Likely unchanged, as concerns about practicality and technical contribution are only partially addressed.

Reviewer kzqx (4→ 4 or 6): Not all concerns are addressed.

---

### Decision · Program_Chairs · 2026-01-26

Reject